# Sparse Deep Learning for Time Series Data: Theory and Applications

**Mingxuan Zhang**
Department of Statistics
Purdue University
zhan3692@purdue.edu

**Yan Sun**
Department of Biostatistics, Epidemiology, and Informatics
University of Pennsylvania
yan.sun@pennmedicine.upenn.edu

**Faming Liang**
Department of Statistics
fmliang@purdue.edu

## Abstract

Sparse deep learning has become a popular technique for improving the performance of deep neural networks in areas such as uncertainty quantification, variable selection, and large-scale network compression. However, most existing research has focused on problems where the observations are independent and identically distributed (i.i.d.), and there has been little work on the problems where the observations are dependent, such as time series data and sequential data in natural language processing. This paper aims to address this gap by studying the theory for sparse deep learning with dependent data. We show that sparse recurrent neural networks (RNNs) can be consistently estimated, and their predictions are asymptotically normally distributed under appropriate assumptions, enabling the prediction uncertainty to be correctly quantified. Our numerical results show that sparse deep learning outperforms state-of-the-art methods, such as conformal predictions, in prediction uncertainty quantification for time series data. Furthermore, our results indicate that the proposed method can consistently identify the autoregressive order for time series data and outperform existing methods in large-scale model compression. Our proposed method has important practical implications in fields such as finance, healthcare, and energy, where both accurate point estimates and prediction uncertainty quantification are of concern.

## 1   Introduction

Over the past decade, deep learning has experienced unparalleled triumphs across a multitude of domains, such as time series forecasting [1, 2, 3, 4, 5], natural language processing [6, 7], and computer vision [8, 9]. However, challenges like generalization and miscalibration [10] persist, posing potential risks in critical applications like medical diagnosis and autonomous vehicles.

In order to enhance the performance of deep neural networks (DNNs), significant research efforts have been dedicated to exploring optimization methods and the loss surface of the DNNs, see, e.g., [11, 12, 13, 14, 15, 16], which have aimed to expedite and direct the convergence of DNNs towards regions that exhibit strong generalization capabilities. While these investigations are valuable, effectively addressing both the challenges of generalization

37th Conference on Neural Information Processing Systems (NeurIPS 2023).

and miscalibration require additional and perhaps more essential aspects: consistent estimation of the underlying input-output mapping and complete knowledge of the asymptotic distribution of predictions. As a highly effective method that addresses both challenges, sparse deep learning has been extensively studied, see, e.g., [17, 18, 19, 20, 21]. Nevertheless, it is important to note that all the studies have been conducted under the assumption of independently and identically distributed (i.i.d.) data. However, in practice, we frequently encounter situations where the data exhibits dependence, such as time series data.

The primary objective of this paper is to address this gap by establishing a theoretical foundation for sparse deep learning with time series data. Specifically, we lay the foundation within the Bayesian framework. For RNNs, by letting their parameters be subject to a mixture Gaussian prior, we establish posterior consistency, structure selection consistency, input-output mapping estimation consistency, and asymptotic normality of predicted values. We validate our theory through numerical experiments on both synthetic and real-world datasets. Our approach outperforms existing state-of-the-art methods in uncertainty quantification and model compression, highlighting its potential for practical applications where both accurate point prediction and prediction uncertainty quantification are of concern.

Additionally, we would elaborate the contribution of this paper in a broader context of statistical modeling. As discussed in [22], two distinct cultures exist for statistical modeling: the 'data modeling culture' and the 'algorithmic modeling culture'. The former focuses on simple generative models that explain the data, potentially lacking a consistent estimate of the true data-generating mechanism due to the model's inherent simplicity. The latter, on the other hand, aims to find models that can predict the data regardless of complexity. Our proposed method occupies a middle ground between these two cultures. It seeks to identify a parsimonious model within the realm of complex models, while also ensuring a consistent estimation of the true data-generating mechanism. From this perspective, this work and related ones, e.g., [17, 18, 19, 20, 21], represent a hybridization of the 'algorithmic modeling culture' and the 'data modeling culture', which holds the potential to expedite advancements in modern data science.

## 2   Related Works

**Sparse deep learning**. Theoretical investigations have been conducted on the approximation power of sparse DNNs across different classes of functions [23, 24]. Recently, [20] has made notable progress by integrating sparse DNNs into the framework of statistical modeling, which offers a fundamentally distinct neural network approximation theory. Unlike traditional theories that lack data involvement and allow connection weights to assume values in an unbounded space to achieve arbitrarily small approximation errors with small networks [25], their theory [20] links network approximation error, network size, and weight bounds to the training sample size. They show that a sparse DNN of size $O(n/\log(n))$ can effectively approximate various types of functions, such as affine and piecewise smooth functions, as $n \to \infty$, where $n$ denotes the training sample size. Additionally, sparse DNNs exhibit several advantageous theoretical guarantees, such as improved interpretability, enabling the consistent identification of relevant variables for high-dimensional nonlinear systems. Building upon this foundation, [21] establishes the asymptotic normality of connection weights and predictions, enabling valid statistical inference for predicting uncertainties. This work extends the sparse deep learning theory of [20, 21] from the case of i.i.d data to the case of time series data.

**Uncertainty quantification**. Conformal Prediction (CP) has emerged as a prominent technique for generating prediction intervals, particularly for black-box models like neural networks. A key advantage of CP is its capability to provide valid prediction intervals for any data distribution, even with finite samples, provided the data meets the condition of exchangeability [26, 27]. While i.i.d. data easily satisfies this condition, dependent data, such as time series, often doesn't. Researchers have extended CP to handle time series data by relying on properties like strong mixing and ergodicity [28, 29]. In a recent work, [30] introduced a random swapping mechanism to address potentially non-exchangeable data, allowing conformal prediction to be applied on top of a model trained with weighted samples. The main focus of this approach was to provide a theoretical basis for the

differences observed in the coverage rate of the proposed method. Another recent study by [31] took a deep dive into the Adaptive Conformal Inference (ACI) [32], leading to the development of the Aggregation Adaptive Conformal Inference (AgACI) method. In situations where a dataset contains a group of similar and independent time series, treating each time series as a separate observation, applying a CP method becomes straightforward [33]. For a comprehensive tutorial on CP methods, one can refer to [34]. Beyond CP, other approaches for addressing uncertainty quantification in time series datasets include multi-horizon probabilistic forecasting [35], methods based on dropout [36], and recursive Bayesian approaches [37].

## 3    Sparse Deep Learning for Time Series Data: Theory

Let $D_n = \{y_1, \ldots, y_n\}$ denote a time series sequence, where $y_i \in \mathbb{R}$. Let $(\Omega, \mathcal{F}, P^*)$ be the probability space of $D_n$, and let $\alpha_k = \sup\{|P^*(y_j \in A, y_{k+j} \in B) - P^*(y_j \in A)P^*(y_{k+j} \in B)| : A, B \in \mathcal{F}, j \in \mathbb{N}^+\}$ be the $k$-th order $\alpha$-mixing coefficient.

**Assumption 3.1.** The time series $D_n$ is (strictly) stationary and $\alpha$-mixing with an exponentially decaying mixing coefficient and follows an autoregressive model of order $l$

$$y_i = \mu^*(y_{i-1:i-l}, \boldsymbol{u}_i) + \eta_i, \tag{1}$$

where $\mu^*$ is a non-linear function, $y_{i-1:i-l} = (y_{i-1}, \ldots, y_{i-l})$, $\boldsymbol{u}_i$ contains optional exogenous variables, and $\eta_i \overset{i.i.d.}{\sim} N(0, \sigma^2)$ with $\sigma^2$ being assumed to be a constant.

*Remark* 3.2. Similar assumptions are commonly adopted to establish asymptotic properties of stochastic processes [38, 39, 40, 41, 42, 29]. For example, the asymptotic normality of the maximum likelihood estimator (MLE) can be established under the assumption that the time series is strictly stationary and ergodic, provided that the model size is fixed [38]. A posterior contraction rate of the autoregressive (AR) model can be obtained by assuming it is $\alpha$-mixing with $\sum_{k=0}^{\infty} a_k^{1-2/s} < \infty$ for some $s > 2$ which is implied by an exponentially decaying mixing coefficient [42]. For stochastic processes that are strictly stationary and $\beta$-mixing, results such as uniform laws of large numbers and convergence rates of the empirical processes [39, 40] can also be obtained.

*Remark* 3.3. Extending the results of [20, 21] to the case that the dataset includes a set of i.i.d. time series, with each time series regarded as an individual observation, is straightforward, this is because all observations are independent and have the same distribution.

### 3.1    Posterior Consistency

Both the MLP and RNN can be used to approximate $\mu^*$ as defined in (1), and for simplicity, we do not explicitly denote the exogenous variables $\boldsymbol{u}_i$ unless it is necessary. For the MLP, we can formulate it as a regression problem, where the input is $\boldsymbol{x}_i = y_{i-1:i-R_l}$ for some $l \leq R_l \ll n$, and the corresponding output is $y_i$, then the dataset $D_n$ can be expressed as $\{(\boldsymbol{x}_i, y_i)\}_{i=1+R_l}^n$. Detailed settings and results for the MLP are given in Appendix B.3. In what follows, we will focus on the RNN, which serves as an extension of the previous studies. For the RNN, we can rewrite the training dataset as $D_n = \{y_{i:i-M_l+1}\}_{i=M_l}^n$ for some $l \leq R_l < M_l \ll n$, i.e., we split the entire sequence into a set of shorter sequences, where $R_l$ denotes an upper bound for the exact AR order $l$, and $M_l$ denotes the length of these shorter sequences (see Figure 1). We assume $R_l$ is known but not $l$ since, in practice, it is unlikely that we know the exact order $l$.

For simplicity of notations, we do not distinguish between weights and biases of the RNN. In this paper, the presence of the subscript $n$ in the notation of a variable indicates its potential to increase with the sample size $n$. To define an RNN with $H_n - 1$ hidden layers, for $h \in \{1, 2, \ldots, H_n\}$, we let $\psi^h$ and $L_h$ denote, respectively, the nonlinear activation function and the number of hidden neurons at layer $h$. We set $L_{H_n} = 1$ and $L_0 = p_n$, where $p_n$ denotes a generic input dimension. Because of the existence of hidden states from the past, the input $\boldsymbol{x}_i$ can contain only $y_{i-1}$ or $y_{i-1:i-r}$ for some $r > 1$. Let $\boldsymbol{w}^h \in \mathbb{R}^{L_h \times L_{h-1}}$ and $\boldsymbol{v}^h \in \mathbb{R}^{L_h \times L_h}$ denote the weight matrices at layer $h$. With these notations, the output of the step $i$ of an RNN model can be expressed as

$$\mu(\boldsymbol{x}_i, \{\boldsymbol{z}_{i-1}^h\}_{h=1}^{H_n-1}, \boldsymbol{\beta}) = \boldsymbol{w}^{H_n} \psi^{H_n-1}[\cdots \psi^1[\boldsymbol{w}^1 \boldsymbol{x}_i + \boldsymbol{v}^1 \boldsymbol{z}_{i-1}^1] \cdots], \tag{2}$$

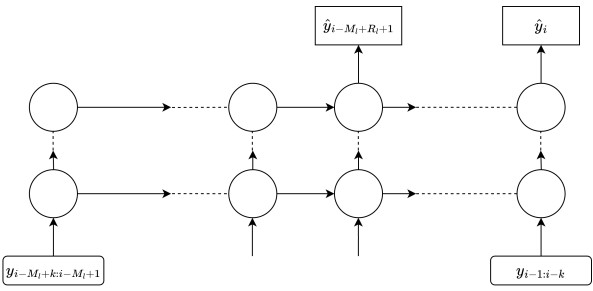

Figure 1: A multi-layer RNN with an input window size of $k$. We restrict the use of the RNN's outputs until the hidden states have accumulated a sufficient quantity of past information to ensure accurate predictions.

where $z_i^h = \psi^h[w^h z_i^{h-1} + v^h z_{i-1}^h]$ denotes the hidden state of layer $h$ at step $i$ with $z_0^h = 0$; and $\beta$ is the collection of all weights, consisting of $K_n = \sum_{h=1}^{H_n}(L_h \times L_{h-1}) + \sum_{h=1}^{H_n-1}(L_h^2) + L_h$ elements. To represent the structure for a sparse RNN, we introduce an indicator variable for each weight in $\beta$. Let $\gamma = \{\gamma_j \in \{0,1\} : j = 1, \ldots, K_n\}$, which specifies the structure of a sparse RNN. To include information on the network structure $\gamma$ and keep the notation concise, we redenote $\mu(x_i, \{z_{i-1}^h\}_{h=1}^{H_n-1}, \beta)$ by $\mu(x_{i:i-M_l+1}, \beta, \gamma)$, as $\{z_{i-1}^h\}_{h=1}^{H_n-1}$ depends only on $(\beta, \gamma)$ and up to $x_{i-1:i-M_l+1}$.

Posterior consistency is an essential concept in Bayesian statistics, which forms the basis of Bayesian inference. While posterior consistency generally holds for low-dimensional problems, establishing it becomes challenging in high-dimensional scenarios. In such cases, the dimensionality often surpasses the sample size, and if the prior is not appropriately elicited, prior information can overpower the data information, leading to posterior inconsistency.

Following [20, 21, 17], we let each connection weight be subject to a mixture Gaussian prior, i.e.,

$$\beta_j \sim \lambda_n N(0, \sigma_{1,n}^2) + (1 - \lambda_n) N(0, \sigma_{0,n}^2), \quad j \in \{1, 2, \ldots, K_n\}, \tag{3}$$

by integrating out the structure information $\gamma$, where $\lambda_n \in (0,1)$ is the mixture proportion, $\sigma_{0,n}^2$ is typically set to a very small number, while $\sigma_{1,n}^2$ is relatively large. Visualizations of the mixture Gaussian priors for different $\lambda_n$, $\sigma_{0,n}^2$, and $\sigma_{1,n}^2$ are given in the Appendix E.

We assume $\mu^*$ can be well approximated by a sparse RNN given enough past information, and refer to this sparse RNN as the true RNN model. To be more specific, we define the true RNN model as

$$(\beta^*, \gamma^*) = \underset{(\beta, \gamma) \in \mathcal{G}_n, \left\| \mu(x_{i:i-M_l+1}, \beta, \gamma) - \mu^*(y_{i-1:i-l}) \right\|_{L_2(\Omega)} \leq \varpi_n}{\arg\min} |\gamma|, \tag{4}$$

where $\mathcal{G}_n := \mathcal{G}(C_0, C_1, \epsilon, p_n, L_1, L_2, \ldots, L_{H_n})$ denotes the space of all valid networks that satisfy the Assumption 3.4 for the given values of $H_n$, $p_n$, and $L_h$'s, and $\varpi_n$ is some sequence converging to 0 as $n \to \infty$. For any given RNN $(\beta, \gamma)$, the error $|\mu^*(\cdot) - \mu(\cdot, \beta, \gamma)|$ can be decomposed as the approximation error $|\mu^*(\cdot) - \mu(\cdot, \beta^*, \gamma^*)|$ and the estimation error $|\mu(\cdot, \beta^*, \gamma^*) - \mu(\cdot, \beta, \gamma)|$. The former is bounded by $\varpi_n$, and the order of the latter will be given in Theorem 3.9. For the sparse RNN, we make the following assumptions:

**Assumption 3.4.** The true sparse RNN model $(\beta^*, \gamma^*)$ satisfies the following conditions:

- The network structure satisfies: $r_n H_n \log n + r_n \log \bar{L} + s_n \log p_n \leq C_0 n^{1-\epsilon}$, where $0 < \epsilon < 1$ is a small constant, $r_n = |\gamma^*|$ denotes the connectivity of $\gamma^*$, $\bar{L} = \max_{1 \leq j \leq H_{n-1}} L_j$ denotes the maximum hidden state dimension, and $s_n$ denotes the input dimension of $\gamma^*$.

- The network weights are polynomially bounded: $\|\beta^*\|_\infty \leq E_n$, where $E_n = n^{C_1}$ for some constant $C_1 > 0$.

*Remark* 3.5. Assumption 3.4 is identical to assumption A.2 of [20], which limits the connectivity of the true RNN model to be of $o(n^{1-\epsilon})$ for some $0 < \epsilon < 1$. Then, as implied by

Lemma 3.10, an RNN of size $O(n/\log(n))$ has been large enough for modeling a time series sequence of length $n$. Refer to [20] for discussions on the universal approximation ability of the neural network under this assumption; the universal approximation ability can still hold for many classes of functions, such as affine function, piecewise smooth function, and bounded $\alpha$-Hölder smooth function.

*Remark* 3.6. The existence of the sparse RNN model stems from Lemma 4.1 of [39], which, through the trick of independent block sequence construction, shows that many properties of the i.i.d processes can be extended to mixing processes. While the lemma was proven for the case of $\beta$-mixing, the author did mention her doubts about its applicability to $\alpha$-mixing. Therefore, at least for the sequences of $\beta$-mixing, which implies $\alpha$-mixing, the non-empty of the sparse RNN set in (4) can be guaranteed for many classes of functions as mentioned in Remark 3.5.

**Assumption 3.7.** The activation function $\psi^h$ is bounded for $h = 1$ (e.g., sigmoid and tanh), and is Lipschitz continuous with a Lipschitz constant of 1 for $2 \leq h \leq H_n$ (e.g., ReLU, sigmoid and tanh).

*Remark* 3.8. Assumption 3.7 mirrors [20, 21], except that we require the activation function for the first layer to be bounded. This extra assumption can be viewed as a replacement for the boundedness assumption for the input variables of a conventional DNN.

Let $d(p_1, p_2)$ denote the integrated Hellinger distance between two conditional densities $p_1(y|\boldsymbol{x})$ and $p_2(y|\boldsymbol{x})$. Let $\pi(\cdot|D_n)$ be the posterior probability of an event. Theorem 3.9 establishes posterior consistency for the RNN model with the mixture Gaussian prior (3).

**Theorem 3.9.** *Suppose Assumptions 3.1, 3.4, and 3.7 hold. If the mixture Gaussian prior (3) satisfies the conditions : $\lambda_n = O(1/[M_l^{H_n} K_n[n^{2M_l H_n}(\bar{L}p_n)]^\tau])$ for some constant $\tau > 0$, $E_n/[H_n \log n + \log \bar{L}]^{1/2} \leq \sigma_{1,n} \leq n^\alpha$ for some constant $\alpha$, and $\sigma_{0,n} \leq \min\{1/[M_l^{H_n}\sqrt{n}K_n(n^{3/2}\sigma_{1,n}/H_n)^{2M_l H_n}], 1/[M_l^{H_n}\sqrt{n}K_n(nE_n/H_n)^{2M_l H_n}]\}$, then there exists an an error sequence $\epsilon_n^2 = O(\varpi_n^2) + O(\zeta_n^2)$ such that $\lim_{n\to\infty} \epsilon_n = 0$ and $\lim_{n\to\infty} n\epsilon_n^2 = \infty$, and the posterior distribution satisfies*

$$\pi(d(p_{\boldsymbol{\beta}}, p_{\mu^*}) \geq C\epsilon_n|D_n) = O_{P^*}(e^{-n\epsilon_n^2}), \tag{5}$$

*for sufficiently large $n$ and $C > 0$, where $\zeta_n^2 = [r_n H_n \log n + r_n \log \bar{L} + s_n \log p_n]/n$, $p_{\mu^*}$ denotes the underlying true data distribution, and $p_{\boldsymbol{\beta}}$ denotes the data distribution reconstructed by the Bayesian RNN based on its posterior samples.*

## 3.2 Uncertainty Quantification with Sparse RNNs

As mentioned previously, posterior consistency forms the basis for Bayesian inference with the RNN model. Based on Theorem 3.9, we further establish structure selection consistency and asymptotic normality of connection weights and predictions for the sparse RNN. In particular, the asymptotic normality of predictions enables the prediction intervals with correct coverage rates to be constructed.

**Structure Selection Consistency** It is known that the neural network often suffers from a non-identifiability issue due to the symmetry of its structure. For instance, one can permute certain hidden nodes or simultaneously change the signs or scales of certain weights while keeping the output of the neural network invariant. To address this issue, we follow [20] to define an equivalent class of RNNs, denoted by $\Theta$, which is a set of RNNs such that any possible RNN for the problem can be represented by one and only one RNN in $\Theta$ via appropriate weight transformations. Let $\nu(\boldsymbol{\beta}, \boldsymbol{\gamma}) \in \Theta$ denote an operator that maps any RNN to $\Theta$. To serve the purpose of structure selection in the space $\Theta$, we consider the marginal posterior inclusion probability (MIPP) approach. Formally, for each connection weight $i = 1, \ldots, K_n$, we define its MIPP as $q_i = \int \sum_{\boldsymbol{\gamma}} e_{i|\nu(\boldsymbol{\beta}, \boldsymbol{\gamma})}\pi(\boldsymbol{\gamma}|\boldsymbol{\beta}, D_n)\pi(\boldsymbol{\beta}|D_n)d\boldsymbol{\beta}$, where $e_{i|\nu(\boldsymbol{\beta}, \boldsymbol{\gamma})}$ is the indicator of $i$ in $\nu(\boldsymbol{\beta}, \boldsymbol{\gamma})$. The MIPP approach selects the connections whose MIPPs exceed a threshold $\hat{q}$. Let $\hat{\boldsymbol{\gamma}}_{\hat{q}} = \{i : q_i > \hat{q}, i = 1, \ldots, K_n\}$ denote an estimator of $\boldsymbol{\gamma}_* = \{i : e_{i|\nu(\boldsymbol{\beta}^*, \boldsymbol{\gamma}^*)} = 1, i = 1, \ldots, K_n\}$. Let $A(\epsilon_n) = \{\boldsymbol{\beta} : d(p_{\boldsymbol{\beta}}, p_{\mu^*}) \leq \epsilon_n\}$ and $\rho(\epsilon_n) = \max_{1 \leq i \leq K_n} \int_{A(\epsilon_n)} \sum_{\boldsymbol{\gamma}} |e_{i|\nu(\boldsymbol{\beta}, \boldsymbol{\gamma})} - e_{i|\nu(\boldsymbol{\beta}^*, \boldsymbol{\gamma}^*)}|\pi(\boldsymbol{\gamma}|\boldsymbol{\beta}, D_n)\pi(\boldsymbol{\beta}|D_n)d\boldsymbol{\beta}$, which measures

the structure difference on $A(\epsilon)$ for the true RNN from those sampled from the posterior. Then we have the following Lemma:

**Lemma 3.10.** *If the conditions of Theorem 3.9 are satisfied and $\rho(\epsilon_n) \to 0$ as $n \to \infty$, then*

(i) $\max_{1 \le i \le K_n}\{|q_i - e_{i|\nu(\boldsymbol{\beta}^*, \boldsymbol{\gamma}^*)}|\} \overset{p}{\to} 0$ *as $n \to \infty$;*

(ii) *(sure screening)* $P(\boldsymbol{\gamma}_* \subset \hat{\boldsymbol{\gamma}}_{\hat{q}}) \overset{p}{\to} 1$ *as $n \to \infty$, for any prespecified $\hat{q} \in (0,1)$;*

(iii) *(consistency)* $P(\boldsymbol{\gamma}_* = \hat{\boldsymbol{\gamma}}_{0.5}) \overset{p}{\to} 1$ *as $n \to \infty$;*

*where $\overset{p}{\to}$ denotes convergence in probability.*

*Remark* 3.11. This lemma implies that we can filter out irrelevant variables and simplify the RNN structure when appropriate. Please refer to Section 5.2 for a numerical illustration.

**Asymptotic Normality of Connection Weights and Predictions** The following two theorems establish the asymptotic normality of $\tilde{\nu}(\boldsymbol{\beta})$ and predictions, where $\tilde{\nu}(\boldsymbol{\beta})$ denotes a transformation of $\boldsymbol{\beta}$ which is invariant with respect to $\mu(\cdot, \boldsymbol{\beta}, \boldsymbol{\gamma})$ while minimizing $\|\tilde{\nu}(\boldsymbol{\beta}) - \boldsymbol{\beta}^*\|_\infty$.

We follow the same definition of asymptotic normality as in [21, 43, 19]. The posterior distribution for the function $g(\boldsymbol{\beta})$ is asymptotically normal with center $g^*$ and variance $G$ if, for $d_{\boldsymbol{\beta}}$ the bounded Lipschitz metric for weak convergence, and $\phi_n$ the mapping $\phi_n : \boldsymbol{\beta} \to \sqrt{n}(g(\boldsymbol{\beta}) - g^*)$, it holds, as $n \to \infty$, that

$$d_{\boldsymbol{\beta}}(\pi(\cdot|D_n) \circ \phi_n^{-1}, N(0, G)) \to 0, \tag{6}$$

in $P^*$-probability, which we also denote $\pi(\cdot|D_n) \circ \phi_n^{-1} \rightsquigarrow N(0, G)$.

The detailed assumptions and setups for the following two theorems are given in Appendix C. For simplicity, we let $M_l = R_l + 1$, and let $l_n(\boldsymbol{\beta}) = \frac{1}{n - R_l} \sum_{i=R_l+1}^{n} \log(p_{\boldsymbol{\beta}}(y_i|y_{i-1:i-R_l}))$ denote the averaged log-likelihood function. Let $H_n(\boldsymbol{\beta})$ denote the Hessian matrix of $l_n(\boldsymbol{\beta})$, and let $h_{i,j}(\boldsymbol{\beta})$ and $h^{i,j}(\boldsymbol{\beta})$ denote the $(i,j)$-th element of $H_n(\boldsymbol{\beta})$ and $H_n^{-1}(\boldsymbol{\beta})$, respectively.

**Theorem 3.12.** *Assume the conditions of Lemma 3.10 hold with $\rho(\epsilon_n) = o(\frac{1}{K_n})$ and additional assumptions hold given in Appendix C, then $\pi(\sqrt{n}(\tilde{\nu}(\boldsymbol{\beta}) - \boldsymbol{\beta}^*)|D_n) \rightsquigarrow N(\boldsymbol{0}, \boldsymbol{V})$ in $P^*$-probability as $n \to \infty$, where $\boldsymbol{V} = (v_{i,j})$, and $v_{i,j} = E(h^{i,j}(\boldsymbol{\beta}^*))$ if $i, j \in \boldsymbol{\gamma}^*$ and $0$ otherwise.*

Theorem 3.13 establishes asymptotic normality of the sparse RNN prediction, which implies prediction consistency and forms the theoretical basis for prediction uncertainty quantification as well.

**Theorem 3.13.** *Assume the conditions of Theorem 3.12 and additional assumptions hold given in Appendix C. Then $\pi(\sqrt{n}(\mu(\cdot, \boldsymbol{\beta}) - \mu(\cdot, \boldsymbol{\beta}^*))|D_n) \rightsquigarrow N(0, \Sigma)$, where $\Sigma = \nabla_{\boldsymbol{\gamma}^*}\mu(\cdot, \boldsymbol{\beta}^*)'H^{-1}\nabla_{\boldsymbol{\gamma}^*}\mu(\cdot, \boldsymbol{\beta}^*)$ and $H = E(-\nabla_{\boldsymbol{\gamma}^*}^2 l_n(\boldsymbol{\beta}^*))$ is the Fisher information matrix.*

## 4 Computation

In the preceding section, we established a theoretical foundation for sparse deep learning with time series data under the Bayesian framework. Building on [20], it is straightforward to show that Bayesian computation can be simplified by invoking the Laplace approximation theorem at the maximum *a posteriori* (MAP) estimator. This essentially transforms the proposed Bayesian method into a regularization method by interpreting the log-prior density function as a penalty for the log-likelihood function in RNN training. Consequently, we can train the regularized RNN model using an optimization algorithm, such as SGD or Adam. To address the local-trap issue potentially suffered by these methods, we train the regularized RNN using a prior annealing algorithm [21], as described in Algorithm 1. For a trained RNN, we sparsify its structure by truncating the weights less than a threshold to zero and further refine the nonzero weights for attaining the MAP estimator. For algorithmic

specifics, refer to Appendix D. Below, we outline the steps for constructing prediction intervals for one-step-ahead forecasts, where $\mu(y_{k-1:k-R_l}, \hat{\boldsymbol{\beta}})$ is of one dimension, and $\hat{\boldsymbol{\beta}}$ and $\hat{\boldsymbol{\gamma}}$ represent the estimators of the network parameters and structure, respectively, as obtained by Algorithm 1:

- Estimate $\sigma^2$ by $\hat{\sigma}^2 = \dfrac{1}{n - R_l - 1} \sum_{i=R_l+1}^{n} \|y_i - \mu(y_{i-1:i-R_l}, \hat{\boldsymbol{\beta}})\|^2$.

- For a test point $y_{k-1:k-R_l}$, estimate $\Sigma$ in Theorem 3.13 by

$$\hat{\varsigma}^2 = \nabla_{\hat{\boldsymbol{\gamma}}} \mu(y_{k-1:k-R_l}, \hat{\boldsymbol{\beta}})'(-\nabla_{\hat{\boldsymbol{\gamma}}}^2 l_n(\hat{\boldsymbol{\beta}}))^{-1} \nabla_{\hat{\boldsymbol{\gamma}}} \mu(y_{k-1:k-R_l}, \hat{\boldsymbol{\beta}}).$$

- The corresponding $(1-\alpha)\%$ prediction interval is given by

$$\mu(y_{k-1:k-R_l}, \hat{\boldsymbol{\beta}}) \pm z_{\alpha/2} \sqrt{\hat{\varsigma}^2/(n - R_l) + \hat{\sigma}^2},$$

where there are $n - R_l$ observations used in training, and $z_{\alpha/2}$ denotes the upper $\alpha/2$-quantile of the standard Gaussian distribution.

For construction of multi-horizon prediction intervals, see Appendix F.

## 5 Numerical Experiments

### 5.1 Uncertainty Quantification

As mentioned in Section 3, we will consider two types of time series datasets: the first type comprises a single time series, and the second type consists of a set of time series. We will compare the performance of our method against the state-of-the-art Conformal Prediction (CP) methods for both types of datasets. We set $\alpha = 0.1$ (the error rate) for all uncertainty quantification experiments in the paper, and so the nominal coverage level of the prediction intervals is 90%.

#### 5.1.1 A Single Time Series: French Electricity Spot Prices

We perform one-step-ahead forecasts on the French electricity spot prices data from 2016 to 2020, which consists of 35,064 observations. A detailed description and visualization of this time series are given in Appendix G.1. Our goal is to predict the 24 hourly prices of the following day, given the prices up until the end of the current day. As the hourly prices exhibit distinct patterns, we fit one model per hour as in the CP baseline [31]. We follow the data splitting strategy used in [31], where the first three years $2016 - 2019$ data are used as the (initial) training set, and the prediction is made for the last year $2019 - 2020$.

For all the methods considered, we use the same underlying neural network model: an MLP with one hidden layer of size 100 and the sigmoid activation function. More details on the training process are provided in Appendix G.1. For the state-of-the-art CP methods, EnbPI-V2 [29], NEX-CP [30], ACI [32] and AgACI [31], we conduct experiments in an online fashion, where the model is trained using a sliding window of the previous three years of data (refer to Figure 4 in the Appendix). Specifically, after constructing the prediction interval for each time step in the prediction period, we add the ground truth value to the training set and then retrain the model with the updated training set. For ACI, we conduct experiments with various values of $\gamma$ and present the one that yields the best performance. In the case of AgACI, we adopt the same aggregation approach as used in [31], namely, the Bernstein Online Aggregation (BOA) method [44] with a gradient trick. We also report the performance of ACI with $\gamma = 0$ as a reference. For NEX-CP, we use the same weights as those employed in their time-series experiments. For EnbPI-V2, we tune the number of bootstrap models and select the one that offers the best performance.

Since this time series exhibits no or minor distribution shift, our method PA is trained in an offline fashion, where the model is fixed for using only the observations between 2016 and 2019, and the observations in the prediction range are used only for final evaluation. That is, our method uses less data information in training compared to the baseline methods.

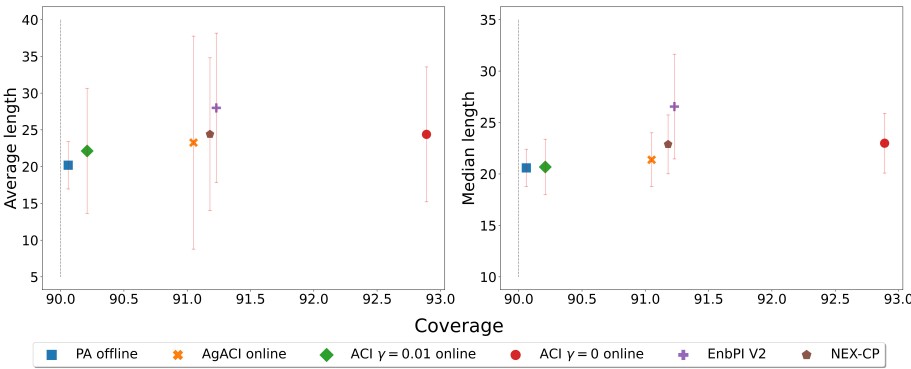

Figure 2: Comparisons of our method and baselines on hourly spot electricity prices in France. **Left**: Average prediction length with vertical error bars indicating the standard deviation of the prediction interval lengths. **Right**: Median prediction length with vertical error bars indicating the interquartile range of the prediction interval lengths. The precise values for these metrics are provided in Table 5 of Appendix G.1.

The results are presented in Figure 2, which includes empirical coverage (with methods that are positioned closer to 90.0 being more effective), median/average prediction interval length, and corresponding interquartile range/standard deviation. As expected, our method is able to train and calibrate the model by using only the initial training set, i.e., the data for 2016-2019, and successfully produces faithful prediction intervals. In contrast, all CP methods produce wider prediction intervals than ours and higher coverage rates than the nominal level of 90%. In addition, ACI is sensitive to the choice of $\gamma$ [31].

### 5.1.2 A Set of Time Series

We conduct experiments using three publicly available real-world datasets: Medical Information Mart for Intensive Care (MIMIC-III), electroencephalography (EEG) data, and daily COVID-19 case numbers within the United Kingdom's local authority districts (COVID-19) [33]. A concise overview of these datasets is presented in Table 1. Our method, denoted as PA-RNN (since the underlying prediction model is an LSTM [45]), is compared to three benchmark methods: CF-RNN [33], MQ-RNN [35], and DP-RNN [36], where LSTM is used for all these methods. To ensure a fair comparison, we adhere to the same model structure, hyperparameters, and data processing steps as specified in [33]. Detailed information regarding the three datasets and training procedures can be found in Appendix G.2.

The numerical results are summarized in Tables 2. Note that the baseline results for EEG and COVID-19 are directly taken from the original paper [33]. We reproduce the baseline results for MIMIC-III, as the specific subset used by [33] is not publicly available. Table 2 indicates that our method consistently outperforms the baselines. In particular, Our method consistently generates shorter prediction intervals compared to the conformal baseline CF-RNN while maintaining the same or even better coverage rate as CF-RNN. Both the MQ-RNN and DP-RNN methods fail to generate prediction intervals that accurately maintain a faithful coverage rate.

Table 1: A brief description of the datasets used in Section 5.1.2, where "Train size" and "Test size" indicate the numbers of training and testing sequences, respectively, and "Length" represents the length of the input sequence.

| Dataset | Train size | Test size | Length | Prediction horizon |
|---------|-----------|-----------|--------|-------------------|
| MIMIC-III [46] | 2692 | 300 | $[3, 47]$ | 2 |
| EEG [47] | 19200 | 19200 | 40 | 10 |
| COVID-19 [48] | 300 | 80 | 100 | 50 |

Table 2: Uncertainty quantification results by different methods for the MIMIC-III, EEG, and COVID-19 data, where "Coverage" represents the joint coverage rate averaged over five different random seeds, the prediction interval length is averaged across prediction horizons and five random seeds, and the numbers in the parentheses indicate the standard deviations (across five random seeds) of the respective means.

| | MIMIC-III | | EEG | | COVID-19 | |
|---|---|---|---|---|---|---|
| **Model** | Coverage | PI length | Coverage | PI length | Coverage | PI length |
| PA-RNN | 90.8%(0.7%) | 2.35(0.26) | 94.69%(0.4%) | 51.02(10.50) | 90.5%(1.4%) | 444.93(203.28) |
| CF-RNN | 92.0%(0.3%) | 3.01(0.17) | 96.5%(0.4%) | 61.86(18.02) | 89.7%(2.4%) | 733.95(582.52) |
| MQ-RNN | 85.3%(0.2%) | 2.64(0.11) | 48.0%(1.8%) | 21.39(2.36) | 15.0%(2.6%) | 136.56(63.32) |
| DP-RNN | 1.2%(0.3%) | 0.12(0.01) | 3.3%(0.3%) | 7.39(0.74) | 0.0%(0.0%) | 61.18(32.37) |

## 5.2 Autoregressive Order Selection

In this section, we evaluate the performance of our method in selecting the autoregressive order for two synthetic autoregressive processes. The first is the non-linear autoregressive (NLAR) process [49, 50, 51]:

$$y_i = -0.17 + 0.85y_{i-1} + 0.14y_{i-2} - 0.31y_{i-3} + 0.08y_{i-7} + 12.80G_1(y_{i-1:i-7}) + 2.44G_2(y_{i-1:i-7}) + \eta_i,$$

where $\eta_i \sim N(0,1)$ represents i.i.d. Gaussian random noises, and the functions $G_1$ and $G_2$ are defined as:

$$G_1(y_{i-1:i-7}) = (1 + \exp\{-0.46(0.29y_{i-1} - 0.87y_{i-2} + 0.40y_{i-7} - 6.68)\})^{-1},$$
$$G_2(y_{i-1:i-7}) = (1 + \exp\{-1.17 \times 10^{-3}(0.83y_{i-1} - 0.53y_{i-2} - 0.18y_{i-7} + 0.38)\})^{-1}.$$

The second is the exponential autoregressive process [52]:

$$y_i = \left(0.8 - 1.1\exp\{-50y_{i-1}^2\}\right)y_{i-1} + \eta_i,$$

where, again, $\eta_i \sim N(0,1)$ denotes i.i.d. Gaussian random noises.

For both synthetic processes, we generate five datasets. Each dataset consists of training, validation, and test sequences. The training sequence has 10000 samples, while the validation and test sequences each contain 1000 samples. For training, we employ a single-layer RNN with a hidden layer width of 1000. Further details on the experimental setting can be found in Appendix G.3.

For the NLAR process, we consider two different window sizes for RNN modeling: 15 (with input as $y_{i-1:i-15}$) and 1 (with input as $y_{i-1}$). Notably, the NLAR process has an order of 7. In the case where the window size is 15, the input information suffices for RNN modeling, rendering the past information conveyed by the hidden states redundant. However, when the window size is 1, this past information becomes indispensable for the RNN. In contrast, the exponential autoregressive process has an order of 1. For all window sizes we explored, namely $1, 3, 5, 7, 10, 15$, the input information is always sufficient for RNN modeling.

We evaluate the predictive performance using mean square prediction error (MSPE) and mean square fitting error (MSFE). The model selection performance is assessed by two metrics: the false selection rate (FSR) and the negative selection rate (NSR). We define $\text{FSR} = \frac{\sum_{j=1}^{5} |\hat{S}_j / S|}{\sum_{j=1}^{5} |\hat{S}_j|}$ and $\text{NSR} = \frac{\sum_{j=1}^{5} |S/\hat{S}_j|}{\sum_{i=j}^{5} |S|}$, where $S$ denotes the set of true variables, and $\hat{S}_j$ represents the set of selected variables for dataset $j$. Furthermore, we provide the final number of nonzero connections for hidden states and the estimated autoregressive orders. The numerical results for the NLAR process are presented in Table 3, while the numerical results for the exponential autoregressive process are given in Table 4.

Our results are promising. Specifically, when the window size is equal to or exceeds the true autoregressive order, all connections associated with the hidden states are pruned, effectively converting the RNN into an MLP. Conversely, if the window size is smaller than the true autoregressive order, a significant number of connections from the hidden states

are retained. Impressively, our method accurately identifies the autoregressive order—a noteworthy achievement considering the inherent dependencies in time series data. Although our method produces a nonzero FSR for the NLAR process, it is quite reasonable considering the relatively short time sequence and the complexity of the functions $G_1$ and $G_2$.

Table 3: Numerical results for the NLAR process: numbers in the parentheses are standard deviations of the respective means, "-" indicates not applicable, ↓ means lower is better, and "#hidden link" denotes the number of nonzero connections from the hidden states. All results are obtained from five independent runs.

| Model | Window size | FSR ↓ | NSR ↓ | AR order | #hidden link | MSPE ↓ | MSFE ↓ |
|---|---|---|---|---|---|---|---|
| PA-RNN | 1 | 0 | 0 | - | 357(21) | 1.056(0.001) | 1.057(0.006) |
| PA-RNN | 15 | 0.23 | 0 | 7.4(0.25) | 0(0) | 1.017(0.008) | 1.020(0.010) |

Table 4: Numerical results for the exponetial autoregressive process.

| Model | Window size | FSR ↓ | NSR ↓ | AR order | #hidden link | MSPE ↓ | MSFE ↓ |
|---|---|---|---|---|---|---|---|
| PA-RNN | 1 | 0 | 0 | 1(0) | 0(0) | 1.004(0.004) | 1.003(0.005) |
| PA-RNN | 3 | 0 | 0 | 1(0) | 0(0) | 1.006(0.005) | 0.999(0.004) |
| PA-RNN | 5 | 0 | 0 | 1(0) | 0(0) | 1.000(0.004) | 1.007(0.005) |
| PA-RNN | 7 | 0 | 0 | 1(0) | 0(0) | 1.006(0.005) | 1.000(0.003) |
| PA-RNN | 10 | 0 | 0 | 1(0) | 0(0) | 1.002(0.004) | 1.002(0.006) |
| PA-RNN | 15 | 0 | 0 | 1(0) | 0(0) | 1.001(0.004) | 1.002(0.007) |

## 5.3   RNN Model Compression

We have also applied our method to RNN model compression, achieving state-of-the-art results. Please refer to Section G.4 in the Appendix for details.

## 6   Discussion

This paper has established the theoretical groundwork for sparse deep learning with time series data, including posterior consistency, structure selection consistency, and asymptotic normality of predictions. Our empirical studies indicate that sparse deep learning can outperform current cutting-edge approaches, such as conformal predictions, in prediction uncertainty quantification. More specifically, compared to conformal methods, our method maintains the same coverage rate, if not better, while generating significantly shorter prediction intervals. Furthermore, our method effectively determines the autoregression order for time series data and surpasses state-of-the-art techniques in large-scale model compression. The theory developed in this paper has included LSTM [45] as a special case, and some numerical examples have been conducted with LSTM; see Section G of the Appendix for the detail. Furthermore, there is room for refining the theoretical study under varying mixing assumptions for time series data, which could broaden applications of the proposed method. Also, the efficacy of the proposed method can potentially be further improved with the elicitation of different prior distributions.

In summary, this paper represents a significant advancement in statistical inference for deep RNNs, which, through sparsing, has successfully integrated the RNNs into the framework of statistical modeling. The superiority of our method over the conformal methods shows further the criticality of consistently approximating the underlying mechanism of the data generation process in uncertainty quantification.

In terms of limitations of the proposed method, one potential concern pertains to the calculation of the inverse of the Fisher information matrix. For large-scale problems, the sparsified model could still retain a large number of non-zero parameters. In such instances, the computational feasibility of calculating the Hessian matrix might become compromised. Nonetheless, an alternative avenue exists in the form of the Bayesian approach, which circumvents the matrix inversion challenge. A concise description of this Bayesian strategy can be found in [21].

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

# Appendix for "Sparse Deep Learning for Time Series Data: Theory and Applications"

## A    Mathematical Facts of Sparse RNNs

For a sparse RNN model with $H_n - 1$ recurrent layers and a single output layer, several mathematical facts can be established. Let's denote the number of nodes in each layer as $L_1, \ldots, L_{H_n}$. Additionally, let $r_{w_i}$ represent the number of non-zero connection weights for $\boldsymbol{w}^i$, and let $r_{v_i}$ denote the number of non-zero connection weights for $\boldsymbol{v}^i$, where $\boldsymbol{w}_i$ and $\boldsymbol{v}_i$ denote the connection weights at two layers of a sparse RNN.

Furthermore, we define $O_{i,j}^t(\boldsymbol{\beta}, \boldsymbol{x}_{t:1})$ as the output value of the $j$-th neuron of the $i$-th layer at time $t$. This output depends on the parameter vector $\boldsymbol{\beta}$ and the input sequence $\boldsymbol{x}_{t:1} = (\boldsymbol{x}_t, \ldots, \boldsymbol{x}_1)$, where $\boldsymbol{x}_t$ represents the generic input to the sparse RNN at time step $t$.

**Lemma A.1.** *Under Assumption 3.7, if a sparse RNN has at most $r_n$ non-zero connection weights (i.e., $r_n = \sum_{i=1}^{H_n} r_{w_i} + \sum_{i=1}^{H_n-1} r_{v_i}$) and $\|\boldsymbol{\beta}\|_\infty = E_n$, then the summation of the absolute outputs of the ith layer at time t is bounded by*

$$\sum_{j=1}^{L_i} |O_{i,j}^t(\boldsymbol{\beta}, \boldsymbol{x}_{t:1})| \leq t^i \left( \prod_{k=1}^{i} r_{w_k} \right) \left( \prod_{k=1}^{i} r_{v_k} \right)^{t-1} (E_n)^{ti}, \quad 1 \leq i \leq H_n - 1,$$

$$\sum_{j=1}^{L_{H_n}} |O_{H_n,j}^t(\boldsymbol{\beta}, \boldsymbol{x}_{t:1})| \leq t^{H_n} \left( \prod_{k=1}^{H_n} r_{w_k} \right) \left( \prod_{k=1}^{H_n-1} r_{v_k} \right)^{t-1} (E_n)^{t(H_n-1)+1}.$$

*Proof.* For simplicity, we rewrite $O_{i,j}^t(\boldsymbol{\beta}, \boldsymbol{x}_{t:1})$ as $O_{i,j}^t$ when appropriate. The lemma is the result from the following facts:

- For $t = 1$ and $1 \leq i \leq H_n$ (Lemma S4 from [20])

$$\sum_{j=1}^{L_i} |O_{i,j}^1| \leq E_n^i \prod_{k=1}^{i} r_{w_k}.$$

- For $t \geq 2$ and $i = 1$ (recursive relationship)

$$\sum_{j=1}^{L_1} |O_{1,j}^t| \leq E_n r_{w_1} + E_n r_{v_1} \sum_{j=1}^{L_1} |O_{1,j}^{t-1}|.$$

- For $t \geq 2$ and $2 \leq i \leq H_n - 1$ (recursive relationship)

$$\sum_{j=1}^{L_i} |O_{i,j}^t| \leq E_n r_{w_i} \sum_{j=1}^{L_{i-1}} |O_{i-1,j}^t| + E_n r_{v_i} \sum_{j=1}^{L_i} |O_{i,j}^{t-1}|.$$

- For $t \geq 2$ and $i = H_n$ (recursive relationship)

$$\sum_{j=1}^{L_{H_n}} |O_{H_n,j}^t| \leq E_n r_{w_{H_n}} \sum_{j=1}^{L_{H_n-1}} |O_{H_n-1,j}^t|.$$

We can verify this lemma by plugging the conclusion into all recursive relationships. We show the steps to verify the case when $t \geq 2$ and $2 \leq i \leq H_n - 1$, since other cases are trivial

to verify.

$$\sum_{j=1}^{L_i} |O_{i,j}^t| \le E_n r_{w_i} \sum_{j=1}^{L_{i-1}} |O_{i-1,j}^t| + E_n r_{v_i} \sum_{j=1}^{L_i} |O_{i,j}^{t-1}|$$

$$\le t^{i-1} \left(\prod_{k=1}^{i} r_{w_k}\right) \left(\prod_{k=1}^{i-1} r_{v_k}\right)^{t-1} (E_n)^{t(i-1)+1}$$

$$+ (t-1)^i \left(\prod_{k=1}^{i} r_{w_k}\right) \left(\prod_{k=1}^{i} r_{v_k}\right)^{t-2} r_{v_i} (E_n)^{(t-1)i+1}$$

$$\le t^{i-1} \left(\prod_{k=1}^{i} r_{w_k}\right) \left(\prod_{k=1}^{i} r_{v_k}\right)^{t-1} (E_n)^{ti} + (t-1)^i \left(\prod_{k=1}^{i} r_{w_k}\right) \left(\prod_{k=1}^{i} r_{v_k}\right)^{t-1} (E_n)^{ti}$$

$$\le t^i \left(\prod_{k=1}^{i} r_{w_k}\right) \left(\prod_{k=1}^{i} r_{v_k}\right)^{t-1} (E_n)^{ti},$$

where the last inequality is due to the fact that

$$\frac{t^{i-1} + (t-1)^i}{t^i} = \frac{1}{t} + \left(\frac{t-1}{t}\right)^i \le \frac{1}{t} + \frac{t-1}{t} = 1.$$

$\square$

**Lemma A.2.** *Under Assumption 3.7, consider two RNNs, $\boldsymbol{\beta}$ and $\widetilde{\boldsymbol{\beta}}$, where the former one is a sparse network satisfying $\|\boldsymbol{\beta}\|_0 = r_n$ and $\|\boldsymbol{\beta}\|_\infty = E_n$, and it's network structure vector is denoted by $\boldsymbol{\gamma}$. Assume that if $|\boldsymbol{\beta}_i - \widetilde{\boldsymbol{\beta}}_i| < \delta_1$ for all $i \in \boldsymbol{\gamma}$ and $|\boldsymbol{\beta}_i - \widetilde{\boldsymbol{\beta}}_i| < \delta_2$ for all $i \notin \boldsymbol{\gamma}$, then*

$$\max_{\boldsymbol{x}_{t:1}} |\mu(\boldsymbol{x}_{t:1}, \boldsymbol{\beta}) - \mu(\boldsymbol{x}_{t:1}, \widetilde{\boldsymbol{\beta}})|$$

$$\le t^{H_n} \delta_1 H_n \left(\prod_{k=1}^{H_n} r_{w_k}\right) (E_n + \delta_1)^{t(H_n+1)-2} \left(\prod_{k=1}^{H_n-1} r_{v_k}\right)^{t-1}$$

$$+ t^{H_n} \delta_2 (p_n L_1 + \sum_{k=1}^{H_n} L_k) \left(\prod_{k=1}^{H_n} [\delta_2 L_k + r_{w_k}(E_n + \delta_1)]\right) \left(\prod_{k=1}^{H_n-1} [\delta_2 L_k + r_{v_k}(E_n + \delta_1)]\right)^{t-1}.$$

*Proof.* Define $\check{\boldsymbol{\beta}}$ such that $\check{\boldsymbol{\beta}}_i = \widetilde{\boldsymbol{\beta}}_i$ for all $i \in \boldsymbol{\gamma}$ and $\check{\boldsymbol{\beta}}_i = 0$ for all $i \notin \boldsymbol{\gamma}$. Let $\check{O}_{i,j}^t$ denote $O_{i,j}^t(\check{\boldsymbol{\beta}}, \boldsymbol{x}_{t:1})$. Then, from the following facts that

- For $t = 1$ and $1 \le i \le H_n$ (Lemma S4 from [20])

$$\sum_{j=1}^{L_i} |\check{O}_{i,j}^1 - O_{i,j}^1| \le i\delta_1(E_n + \delta_1)^{i-1} \left(\prod_{k=1}^{i} r_{w_k}\right).$$

- For $t \ge 2$ and $i = 1$ (recursive relationship)

$$\sum_{j=1}^{L_1} |\check{O}_{1,j}^t - O_{1,j}^t| \le r_{w_1}\delta_1 + r_{v_1}\delta_1 \sum_{k=1}^{L_1} |O_{1,k}^{t-1}| + r_{v_1}(E_n + \delta_1) \sum_{k=1}^{L_1} |\check{O}_{1,k}^{t-1} - O_{1,k}^{t-1}|.$$

- For $t \ge 2$ and $2 \le i \le H_n - 1$ (recursive relationship)

$$\sum_{j=1}^{L_i} |\check{O}_{i,j}^t - O_{i,j}^t| \le r_{w_i}\delta_1 \sum_{k=1}^{L_{i-1}} |O_{i-1,k}^t| + r_{w_i}(E_n + \delta_1) \sum_{k=1}^{L_{i-1}} |\check{O}_{i-1,k}^t - O_{i-1,k}^t|$$

$$+ r_{v_i}\delta_1 \sum_{k=1}^{L_i} |O_{i,k}^{t-1}| + r_{v_i}(E_n + \delta_1) \sum_{k=1}^{L_i} |\check{O}_{i,k}^{t-1} - O_{i,k}^{t-1}|.$$

- For $t \geq 2$ and $i = H_n$ (recursive relationship)

$$\sum_{j=1}^{L_{H_n}} |\check{O}_{H_n,j}^t - O_{H_n,j}^t| \leq r_{w_{H_n}} \delta_1 \sum_{k=1}^{L_{H_n-1}} |O_{H_n-1,k}^t|$$

$$+ r_{w_{H_n}}(E_n + \delta_1) \sum_{k=1}^{L_{H_n-1}} |\check{O}_{H_n-1,k}^t - O_{H_n-1,k}^t|.$$

We have

- For $1 \leq i \leq H_n - 1$

$$\sum_{j=1}^{L_i} |\check{O}_{i,j}^t - O_{i,j}^t| \leq t^{i+1} i \delta_1 \left( \prod_{k=1}^i r_{w_k} \right) \left( \prod_{k=1}^i r_{v_k} \right)^{t-1} (E_n + \delta_1)^{(i+1)t-2}.$$

- For $i = H_n$

$$|\mu(\boldsymbol{x}_{t:1}, \boldsymbol{\beta}) - |\mu(\boldsymbol{x}_{t:1}, \check{\boldsymbol{\beta}})| = \sum_{j=1}^{L_{H_n}} |\check{O}_{H_n,j}^t - O_{H_n,j}^t|$$

$$\leq t^{H_n} H_n \delta_1 \left( \prod_{k=1}^{H_n} r_{w_k} \right) \left( \prod_{k=1}^{H_n-1} r_{v_k} \right)^{t-1} (E_n + \delta_1)^{(H_n+1)t-2}.$$

We can verify the above conclusion by plugging it into all recursive relationships. We show the steps to verify the case when $t \geq 2$ and $2 \leq i \leq H_n - 1$, since other cases are trivial to

verify.

$$\sum_{j=1}^{L_i} |\check{O}_{i,j}^t - O_{i,j}^t| \le r_{w_i}\delta_1 \sum_{k=1}^{L_{i-1}} |O_{i-1,k}^t| + r_{w_i}(E_n + \delta_1) \sum_{k=1}^{L_{i-1}} |\check{O}_{i-1,k}^t - O_{i-1,k}^t|$$

$$+ r_{v_i}\delta_1 \sum_{k=1}^{L_i} |O_{i,k}^{t-1}| + r_{v_i}(E_n + \delta_1) \sum_{k=1}^{L_i} |\check{O}_{i,k}^{t-1} - O_{i,k}^{t-1}|$$

$$\le \delta_1 t^{i-1} \left(\prod_{k=1}^{i} r_{w_k}\right) \left(\prod_{k=1}^{i-1} r_{v_k}\right)^{t-1} (E_n + \delta_1)^{ti-t}$$

$$+ \delta_1 t^i (i-1) \left(\prod_{k=1}^{i} r_{w_k}\right) \left(\prod_{k=1}^{i-1} r_{v_k}\right)^{t-1} (E_n + \delta_1)^{ti-1}$$

$$+ \delta_1 (t-1)^i \left(\prod_{k=1}^{i} r_{w_k}\right) \left(\prod_{k=1}^{i} r_{v_k}\right)^{t-2} r_{v_i}(E_n + \delta_1)^{ti-i}$$

$$+ \delta_1 (t-1)^{i+1} i \left(\prod_{k=1}^{i} r_{w_k}\right) \left(\prod_{k=1}^{i} r_{v_k}\right)^{t-2} r_{v_i}(E_n + \delta_1)^{ti-i+t-2}$$

$$\le \delta_1 t^{i-1} \left(\prod_{k=1}^{i} r_{w_k}\right) \left(\prod_{k=1}^{i} r_{v_k}\right)^{t-1} (E_n + \delta_1)^{(i+1)t-2}$$

$$+ \delta_1 t^i (i-1) \left(\prod_{k=1}^{i} r_{w_k}\right) \left(\prod_{k=1}^{i} r_{v_k}\right)^{t-1} (E_n + \delta_1)^{(i+1)t-2}$$

$$+ \delta_1 (t-1)^i \left(\prod_{k=1}^{i} r_{w_k}\right) \left(\prod_{k=1}^{i} r_{v_k}\right)^{t-1} (E_n + \delta_1)^{(i+1)t-2}$$

$$+ \delta_1 (t-1)^{i+1} i \left(\prod_{k=1}^{i} r_{w_k}\right) \left(\prod_{k=1}^{i} r_{v_k}\right)^{t-1} (E_n + \delta_1)^{(i+1)t-2}$$

$$\le t^{i+1} i \delta_1 \left(\prod_{k=1}^{i} r_{w_k}\right) \left(\prod_{k=1}^{i} r_{v_k}\right)^{t-1} (E_n + \delta_1)^{(i+1)t-2},$$

where the last inequality is due to the fact that for $t \ge 2$ and $i \ge 2$, it is easy to see that

$$\frac{t^{i-1} + (i-1)t^i + (t-1)^i + i(t-1)^{i+1}}{it^{i+1}} = \frac{1}{it^2} + \frac{i-1}{i}\frac{1}{t} + \frac{1}{it}\left(\frac{t-1}{t}\right)^i + \left(\frac{t-1}{t}\right)^{i+1}$$

$$\le \frac{1}{2t^2} + \frac{1}{t} + \frac{(t-1)^2}{2t^3} + \left(\frac{t-1}{t}\right)^3 = \frac{t + 2t^2 + t^2 - 2t + 1 + 2(t-1)^3}{2t^3}$$

$$= 1 + \frac{-3t^2 + 5t - 1}{2t^3} \le 1.$$

Now let $\widetilde{O}_{i,j}^t$ denotes $O_{i,j}^t(\widetilde{\boldsymbol{\beta}}, \boldsymbol{x}_{1:t})$, then from the facts that

- For $1 \le t \le T$ and $1 \le i \le H_n$

$$\sum_{j=1}^{L_i} |\check{O}_{i,j}^t| \le t^i \left(\prod_{k=1}^{i} r_{w_k}\right) \left(\prod_{k=1}^{i} r_{v_k}\right)^{t-1} (E_n + \delta_1)^{ti}, \quad 1 \le i \le H_n - 1, \ 1 \le t \le T,$$

$$\sum_{j=1}^{L_{H_n}} |\check{O}_{H_n,j}^t| \le t^{H_n} \left(\prod_{k=1}^{H_n} r_{w_k}\right) \left(\prod_{k=1}^{H_n-1} r_{v_k}\right)^{t-1} (E_n + \delta_1)^{t(H_n-1)+1}, \quad 1 \le t \le T.$$

This can be easily derived by Lemma A.1 and the fact that $\left\|\check{\boldsymbol{\beta}}\right\|_\infty \leq E_n + \delta_1$.

- For $t = 1$ and $1 \leq i \leq H_n$ (Lemma S4 from [20])

$$\sum_{j=1}^{L_i} |\check{O}_{i,j}^1 - \widetilde{O}_{i,j}^1| \leq \delta_2\left(p_n L_1 + \sum_{k=1}^{i} L_k\right)\left(\prod_{k=1}^{i}[\delta_2 L_k + r_{w_k}(E_n + \delta_1)]\right).$$

- For $t \geq 2$ and $i = 1$ (recursive relationship)

$$\sum_{j=1}^{L_1} |\check{O}_{1,j}^t - \widetilde{O}_{1,j}^t| \leq \delta_2 L_1 p_n + \delta_2 L_1 \sum_{k=1}^{L_1} |\check{O}_{1,k}^{t-1} - \widetilde{O}_{1,k}^{t-1}| + r_{v_1}(E_n + \delta_1)\sum_{k=1}^{L_1} |\check{O}_{1,k}^{t-1} - \widetilde{O}_{1,k}^{t-1}|$$
$$+ \delta_2 L_1 \sum_{k=1}^{L_1} |\check{O}_{1,k}^{t-1}|.$$

- For $t \geq 2$ and $2 \leq i \leq H_n - 1$ (recursive relationship)

$$\sum_{j=1}^{L_i} |\check{O}_{i,j}^t - \widetilde{O}_{i,j}^t| \leq \delta_2 L_i \sum_{k=1}^{L_{i-1}} |\check{O}_{i-1,k}^t - \widetilde{O}_{i-1,k}^t| + r_{w_i}(E_n + \delta_1)\sum_{k=1}^{L_{i-1}} |\check{O}_{i-1,k}^t - \widetilde{O}_{i-1,k}^t|$$
$$+ \delta_2 L_i \sum_{k=1}^{L_{i-1}} |\check{O}_{i-1,k}^t| + \delta_2 L_i \sum_{k=1}^{L_i} |\check{O}_{i,k}^{t-1} - \widetilde{O}_{i,k}^{t-1}|$$
$$+ r_{v_i}(E_n + \delta_1)\sum_{k=1}^{L_i} |\check{O}_{i,k}^{t-1} - \widetilde{O}_{i,k}^{t-1}| + \delta_2 L_i \sum_{k=1}^{L_i} |\check{O}_{i,k}^{t-1}|.$$

- For $t \geq 2$ and $i = H_n$ (recursive relationship)

$$\sum_{j=1}^{L_{H_n}} |\check{O}_{H_n,j}^t - \widetilde{O}_{H_n,j}^t| \leq \delta_2 L_{H_n} \sum_{k=1}^{L_{H_n-1}} |\check{O}_{H_n-1,k}^t - \widetilde{O}_{H_n-1,k}^t|$$
$$+ r_{w_{H_n}}(E_n + \delta_1)\sum_{k=1}^{L_{H_n-1}} |\check{O}_{H_n-1,k}^t - \widetilde{O}_{H_n-1,k}^t|$$
$$+ \delta_2 L_{H_n} \sum_{k=1}^{L_{H_n-1}} |\check{O}_{H_n-1,k}^t|.$$

We have

- For $1 \leq i \leq H_n - 1$

$$\sum_{j=1}^{L_i} |\check{O}_{i,j}^t - \widetilde{O}_{i,j}^t| \leq t^{i+1}\delta_2\left(p_n L_1 + \sum_{k=1}^{i} L_k\right)\left(\prod_{k=1}^{i}[\delta_2 L_k + r_{w_k}(E_n + \delta_1)]\right)$$
$$\times \left(\prod_{k=1}^{i}[\delta_2 L_k + r_{v_k}(E_n + \delta_1)]\right)^{t-1}.$$

- For $i = H_n$

$$|\mu(\boldsymbol{x}_{t:1}, \check{\boldsymbol{\beta}}) - \mu(\boldsymbol{x}_{t:1}, \widetilde{\boldsymbol{\beta}})| = \sum_{j=1}^{L_{H_n}} |\check{O}_{H_n,j}^t - O_{H_n,j}^t|$$
$$\leq t^{H_n}\delta_2\left(p_n L_1 + \sum_{k=1}^{H_n} L_k\right)\left(\prod_{k=1}^{H_n}[\delta_2 L_k + r_{w_k}(E_n + \delta_1)]\right)\left(\prod_{k=1}^{H_n-1}[\delta_2 L_k + r_{v_k}(E_n + \delta_1)]\right)^{t-1}.$$

We can verify the above conclusion in a similar approach. The proof is completed by summation of the bound for $|\mu(\boldsymbol{x}_{t:1}, \boldsymbol{\beta}) - \mu(\boldsymbol{x}_{t:1}, \check{\boldsymbol{\beta}})|$ and $|\mu(\boldsymbol{x}_{t:1}, \check{\boldsymbol{\beta}}) - \mu(\boldsymbol{x}_{t:1}, \widetilde{\boldsymbol{\beta}})|$. $\qquad\square$

# B    Proofs on Posterior Consistency: A Single Time Series

To establish posterior consistency for DNNs with i.i.d. data, [20] utilized Proposition 1 from [53]. This lemma provides three sufficient conditions for proving posterior consistency for general statistical models with i.i.d. data, along with a posterior contraction rate. In this paper, we aim to establish posterior consistency for DNNs with stochastic processes, specifically time series, that are strictly stationary and $\alpha$-mixing with an exponentially decaying mixing coefficient [54, 42].

Consider a time series $D_n = (z_1, z_2, \ldots, z_n)$ defined on a probability space $(\Omega, \mathcal{F}, P^*)$, which satisfies the assumptions outlined in Assumption 1. For simplicity, we assume that the initial values $z_{1-l}, \ldots, z_0$ are fixed and given.

Let $\mathbb{P}_n$ denote a set of probability densities, let $\mathbb{P}_n^c$ denote the complement of $\mathbb{P}_n$, and let $\epsilon_n$ denote a sequence of positive numbers. Let $N(\epsilon_n, \mathbb{P}_n)$ be the minimum number of Hellinger balls of radius $\epsilon_n$ that are needed to cover $\mathbb{P}_n$, i.e., $N(\epsilon_n, \mathbb{P}_n)$ is the minimum of all number $k$'s such that there exist sets $S_j = \{p : d(p, p_j) \leq \epsilon_n\}, j = 1, ..., k$, with $P_n \subset \cup_{j=1}^k S_j$ holding, where

$$d(p, q) = \sqrt{\int \int \int (\sqrt{p(z|\boldsymbol{x})} - \sqrt{q(z|\boldsymbol{x})})^2 v(\boldsymbol{x}) dz d\boldsymbol{x}},$$

denotes the integrated Hellinger distance [42, 54] between the two conditional densities $p(z|\boldsymbol{x})$ and $q(z|\boldsymbol{x})$, where $\boldsymbol{x} \in \mathbb{R}^l$ contains the history up to $l$ time steps of $z$, and $v(\boldsymbol{x})$ is the probability density function of the marginal distribution of $\boldsymbol{x}$. Note that $v$ is invariant with respect to time index $i$ due to the strictly stationary assumption.

For $D_n$, denote the corresponding true conditional density by $p^*$. Define $\pi(\cdot)$ as the prior density, and $\pi(\cdot|D_n)$ as the posterior. Define $\hat{\pi}(\epsilon) = \pi[d(p, p^*) > \epsilon|D_n]$ for each $\epsilon > 0$. Assume the conditions:

  (a) $\log N(\epsilon_n, \mathbb{P}_n) \leq n\epsilon_n^2$ for all sufficiently large $n$.

  (b) $\pi(\mathbb{P}_n^c) \leq e^{-bn\epsilon_n^2}$ for some $b > 0$ and all sufficiently large $n$.

  (c) Let $\|f\|_s = (\int |f|^s dr)^{1/s}$, then $\pi(\|p - p^*\|_s \leq b'\epsilon_n) \geq e^{-\gamma n\epsilon_n^2}$ for some $b', \gamma > 0$, $s > 2$, and all sufficiently large $n$.

**Lemma B.1.** *Under the conditions (a), (b) and (c), given sufficiently large $n$, $\lim_{n \to \infty} \epsilon_n = 0$, and $\lim_{n \to \infty} n\epsilon_n^2 = \infty$, we have for some large $C > 0$,*

$$\hat{\pi}(C\epsilon_n) = \pi(d(p, p^*) \geq C\epsilon_n|D_n) = O_{P^*}(e^{-n\epsilon_n^2}). \tag{7}$$

*Proof.* This Lemma can be proved with similar arguments used in Section 9.5.3, Theorem 8.19, and Theorem 8.29 of [54]. □

## B.1    Posterior Consistency with a General Shrinkage Prior

Let $\boldsymbol{\beta}$ denote the vector of all connection weights of a RNN. To prove Theorem 3.9, we first consider a general shrinkage prior that all entries of $\boldsymbol{\beta}$ are subject to an independent continuous prior $\pi_b$, i.e., $\pi(\boldsymbol{\beta}) = \prod_{j=1}^{K_n} \pi_b(\beta_j)$, where $K_n$ denotes the total number of elements of $\boldsymbol{\beta}$. Theorem B.2 provides a sufficient condition for posterior consistency.

**Theorem B.2.** *(Posterior consistency) Suppose Assumptions 3.1 - 3.7 hold. If the prior $\pi_b$ satisfies that*

$$\log(1/\underline{\pi}_b) = O(H_n \log n + \log \bar{L}), \tag{8}$$

$$\pi_b\{[-\eta_n, \eta_n]\} \geq 1 - \frac{1}{K_n} \exp\{-\tau[H_n \log n + \log \bar{L} + \log p_n]\}, \tag{9}$$

$$\pi_b\{[-\eta_n', \eta_n']\} \geq 1 - \frac{1}{K_n}, \tag{10}$$

$$-\log[K_n \pi_b(|\beta_j| > M_n)] \succ n\epsilon_n^2, \tag{11}$$

*for some $\tau > 0$, where*

$$\eta_n < 1/[\sqrt{n}M_l^{H_n}K_n(n/H_n)^{2M_lH_n}(c_0M_n)^{2M_lH_n}],$$
$$\eta_n' < 1/[\sqrt{n}M_l^{H_n}K_n(r_n/H_n)^{2M_lH_n}(c_0E_n)^{2M_lH_n}],$$

*with some $c_0 > 1$, $\underline{\pi}_b$ is the minimal density value of $\pi_b$ within the interval $[-E_n - 1, E_n + 1]$, and $M_n$ is some sequence satisfying $\log(M_n) = O(\log(n))$. Then, there exists a sequence $\epsilon_n$, satisfying $n\epsilon_n^2 \asymp r_nH_n\log n + r_n\log\bar{L} + s_n\log p_n + n\varpi_n^2$ and $\epsilon_n \prec 1$, such that*

$$\hat{\pi}(M\epsilon_n) = \pi(d(p, p^*) \geq M\epsilon_n|D_n) = O_{P^*}(e^{-n\epsilon_n^2})$$

*for some large $M > 0$.*

*Proof.* To prove this theorem, it suffices to check all three conditions listed in Lemma B.1

**Checking condition (c):**

Consider the set $A = \{\boldsymbol{\beta} : \max_{j\in\gamma^*}|\beta_j - \beta_j^*| \leq \omega_n, \max_{j\notin\gamma^*}|\beta_j - \beta_j^*| \leq \omega_n'\}$, where

$$\omega_n \leq \frac{c_1\epsilon_n}{M_l^{H_n}H_n(r_n/H_n)^{2M_lH_n}(c_0E_n)^{2M_lH_n}},$$
$$\omega_n' \leq \frac{c_1\epsilon_n}{M_l^{H_n}K_n(r_n/H_n)^{2M_lH_n}(c_0E_n)^{2M_lH_n}},$$

for some $c_1 > 0$ and $c_0 > 1$. If $\boldsymbol{\beta} \in A$, by Lemma A.2, we have

$$|\mu(\boldsymbol{x}_{M_l:1}, \boldsymbol{\beta}) - \mu(\boldsymbol{x}_{M_l:1}, \boldsymbol{\beta}^*)| \leq 3c_1\epsilon_n.$$

By the definition (4), we have

$$|\mu(\boldsymbol{x}_{M_l:1}, \boldsymbol{\beta}) - \mu^*(\boldsymbol{x}_{M_l:1})| \leq 3c_1\epsilon_n + \varpi_n$$

Finally for some $s > 2$ (for simplicity, we take $s$ to be an even integer),

$$\|p_{\boldsymbol{\beta}} - p_{\mu^*}\|_s = \left(\int (|\mu(\boldsymbol{x}_{M_l:1}, \boldsymbol{\beta}) - \mu^*(\boldsymbol{x}_{M_l:1})|)^s v(\boldsymbol{x}_{M_l:1})d\boldsymbol{x}_{M_l:1}\right)^{1/s}$$
$$\leq \left(\int (3c_1\epsilon_n + \varpi_n)^s v(\boldsymbol{x}_{M_l:1})d\boldsymbol{x}_{M_l:1}\right)^{1/s}$$
$$\leq 3c_1\epsilon_n + \varpi_n.$$

For any small $b' > 0$, condition (c) is satisfied as long as $c_1$ is sufficiently small, $\epsilon_n \geq C_0\varpi_n$ for some large $C_0$, and the prior satisfies $-\log\pi(A) \leq \gamma n\epsilon_n^2$. Since

$$\pi(A) \geq (\pi_b([E_n - \omega_n, E_n + \omega_n]))^{r_n} \times \pi(\{\max_{j\notin\gamma^*}|\beta_j| \leq \omega_n'\})$$
$$\geq (2\underline{\pi}_b\omega_n)^{r_n} \times \pi_b([-\omega_n', \omega_n'])^{K_n - r_n}$$
$$\geq (2\underline{\pi}_b\omega_n)^{r_n}(1 - 1/K_n)^{K_n},$$

where the last inequality is due to the fact that $\omega_n' \gg \eta_n'$. Note that $\lim_{n\to\infty}(1 - 1/K_n)^{K_n} = e^{-1}$. Since $\log(1/\omega_n) \asymp 2M_lH_n\log(E_n) + 2M_lH_n\log(r_n/H_n) + \log(1/\epsilon_n) + \text{constant} = O(H_n\log n)$ (note that $\log(1/\epsilon_n) = O(\log n)$), then for sufficiently large $n$,

$$-\log\pi(A) \leq r_n\log\left(\frac{1}{\underline{\pi}_b}\right) + r_nO(H_n\log(n)) + r_n\log\left(\frac{1}{2}\right) + 1$$
$$= O(r_nH_n\log(n) + r_n\log\bar{L}).$$

Thus, the prior satisfies $-\log\pi(A) \leq \gamma n\epsilon_n^2$ for sufficiently large $n$, when $n\epsilon_n^2 \geq C_0(r_nH_n\log n + r_n\log\bar{L})$ for some sufficiently large constant $C_0$. Thus condition (c) holds.

**Checking condition (a):**

Let $\mathbb{P}_n$ denote the set of probability densities for the RNNs whose parameter vectors satisfy

$$\boldsymbol{\beta} \in B_n = \left\{|\beta_j| \leq M_n, \Gamma_{\boldsymbol{\beta}} = \{i : |\beta_i| \geq \delta_n'\} \text{ satisfies } |\Gamma_{\boldsymbol{\beta}}| \leq k_nr_n, |\Gamma_{\boldsymbol{\beta}}|_{in} \leq k_n's_n\}\right\},$$

where $|\Gamma_{\boldsymbol{\beta}}|_{in}$ denotes the number of input connections with the absolute weights greater than $\delta'_n$, $k_n (\leq n/r_n)$ and $k'_n (\leq n/s_n)$ will be specified later, and

$$\delta'_n = \frac{c_1 \epsilon_n}{M_l^{H_n} K_n (k_n r_n / H_n)^{2 M_l H_n} (c_0 M_n)^{2 M_l H_n}},$$

for some $c_1 > 0, c_0 > 0$. Let

$$\delta_n = \frac{c_1 \epsilon_n}{M_l^{H_n} H_n (k_n r_n / H_n)^{2 M_l H_n} (c_0 M_n)^{2 M_l H_n}}.$$

Consider two parameter vectors $\boldsymbol{\beta}^u$ and $\boldsymbol{\beta}^v$ in set $B_n$, such that there exists a structure $\boldsymbol{\gamma}$ with $|\boldsymbol{\gamma}| \leq k_n r_n$ and $|\boldsymbol{\gamma}|_{in} \leq k'_n s_n$, and $|\boldsymbol{\beta}^u_j - \boldsymbol{\beta}^v_j| \leq \delta_n$ for all $j \in \boldsymbol{\gamma}$, $\max(|\boldsymbol{\beta}^v_j|, |\boldsymbol{\beta}^u_j|) \leq \delta'_n$ for all $j \notin \boldsymbol{\gamma}$. Hence, by Lemma A.2, we have that $|\mu(\boldsymbol{x}_{1:M_l}, \boldsymbol{\beta}^u) - \mu(\boldsymbol{x}_{1:M_l}, \boldsymbol{\beta}^v)|^2 \leq 9 c_1^2 \epsilon_n^2$. For two normal distributions $N(\mu_1, \sigma^2)$ and $N(\mu_2, \sigma^2)$, define the corresponding Kullback-Leibler divergence as

$$d_0(p_{\mu_1}, p_{\mu_2}) = \frac{1}{2\sigma^2} |\mu_2 - \mu_1|^2.$$

Together with the fact that $2d(p_{\mu_1}, p_{\mu_2})^2 \leq d_0(p_{\mu_1}, p_{\mu_2})$, we have

$$d(p_{\boldsymbol{\beta}^u}, p_{\boldsymbol{\beta}^v}) \leq \sqrt{d_0(p_{\boldsymbol{\beta}^u}, p_{\boldsymbol{\beta}^v})} \leq \sqrt{C(9 + o(1)) c_1^2 \epsilon_n^2} \leq \epsilon_n$$

for some $C > 0$, given a sufficiently small $c_1$.

Given the above results, one can bound the packing number $N(\mathbb{P}_n, \epsilon_n)$ by $\sum_{j=1}^{k_n r_n} \chi_{H_n}^j (\frac{2M_n}{\delta_n})^j$ where $\chi_{H_n}^j$ denotes the number of all valid networks who has exact $j$ connections and has no more than $k'_n s_n$ inputs. Since $\log \chi_{H_n}^j \leq k'_n s_n \log p_n + j \log(k' s_n L_1 + 2 H_n \bar{L}^2)$, $\log M_n = O(\log n)$, $k_n r_n \leq n$, and $k'_n s_n \leq n$, then

$$\begin{aligned}
\log N(\mathbb{P}_n, \epsilon_n) &\leq \log \left( k_n r_n \chi_{H_n}^{k_n r_n} (\frac{2M_n}{\delta_n})^{k_n r_n} \right) \\
&\leq \log k_n r_n + k'_n s_n \log p_n + k_n r_n \log(2 H_n) + 2 k_n r_n \log(\bar{L} + k'_n s_n) \\
&\quad + k_n r_n \log \frac{2 M_n M_l^{H_n} H_n (k_n r_n / H_n)^{2 M_l H_n} (c_0 M_n)^{2 M_l H_n}}{c_1 \epsilon_n} \\
&= k_n r_n O(H_n \log n + \log \bar{L}) + k'_n s_n \log p_n.
\end{aligned}$$

We can choose $k_n$ and $k'_n$ such that for sufficiently large $n$, $k_n r_n \{H_n \log n + \log \bar{L}\} \asymp k'_n s_n \log p_n \asymp n \epsilon_n^2$ and then $\log N(\mathbb{P}_n, \epsilon_n) \leq n \epsilon_n^2$.

**Checking condition (b):**

**Lemma B.3.** *(Theorem 1 of [55]) Let $X \sim Binomial(n, p)$ be a Binomial random variable. For any $1 < k < n - 1$*

$$Pr(X \geq k + 1) \leq 1 - \Phi(sign(k - np) [2 n H(p, k/n)]^{1/2}),$$

*where $\Phi$ is the cumulative distribution function (CDF) of the standard Gaussian distribution and $H(p, k/n) = (k/n) \log(k/np) + (1 - k/n) \log[(1 - k/n)/(1 - p)]$.*

Now,

$$\pi(\mathbb{P}_n^c) \leq Pr(Binomial(K_n, v_n) > k_n r_n) + K_n \pi_b(|\beta_j| > M_n) + Pr(|\Gamma_{\boldsymbol{\beta}}|_{in} > k'_n s_n),$$

where $v_n = 1 - \pi_b([-\delta'_n, \delta'_n])$. By the condition on $\pi_b$ and the fact that $\delta'_n \gg \eta'_n$, it is easy to see that $v_n \leq \exp\{\tau[H_n \log n + \log \bar{L} + \log p_n] - \log K_n\}$ for some constant $\tau > 0$. Thus, by Lemma B.3, $-\log Pr(Binomial(K_n, v_n) > k_n r_n) \approx \tau k_n r_n [H_n \log n + \log \bar{L} + \log p_n] \gtrsim n \epsilon_n^2$ due to the choice of $k_n$, and $-\log Pr(|\Gamma_{\boldsymbol{\beta}}|_{in} \geq k'_n s_n) \approx k'_n s_n [\tau(H_n \log n + \log \bar{L} + \log p_n) + \log(p_n K_n / L_1)] \gtrsim n \epsilon_n^2$ due to the choice of $k'_n$. Thus, condition (b) holds as well. Then by lemma B.1, the proof is completed. $\qquad\square$

## B.2 Proof of Theorem 3.9

*Proof.* To prove Theorem 3.9 in the main text, it suffices to verify the four conditions on $\pi_b$ listed in Theorem B.2. Let $M_n = \max(\sqrt{2n}\sigma_{1,n}, E_n)$. Condition 8 can be verified by choosing $\sigma_{1.n}$ such that $E_n^2/2\sigma_{1.n}^2 + \log \sigma_{1,n}^2 = O(H_n \log n + \log \bar{L})$. Conditions 9 and 10 can be verified by setting $\lambda_n = 1/\{M_l^{H_n} K_n [n^{2M_l H_n}(\bar{L}p_n)]^\tau\}$ and $\sigma_{0,n} \prec 1/\{M_l^{H_n}\sqrt{n}K_n(n/H_n)^{2M_l H_n}(c_0 M_n)^{2M_l H_n}\}$. Finally, condition 11 can be verified by $M_n \geq 2n\sigma_{0.n}^2$ and $\tau[H_n \log n + \log \bar{L} + \log p_n] + M_n^2/2\sigma_{1.n}^2 \geq n$. Finally, based on the proof above we see that $n\epsilon_n^2 \asymp r_n H_n \log n + r_n \log \bar{L} + s_n \log p_n + n\varpi_n^2$ and $\lim_{n\to\infty} \epsilon_n = 0$. $\square$

## B.3 Posterior Consistency for Multilayer Perceptrons

As highlighted in Section 3, the MLP can be formulated as a regression problem. Here, the input is $\boldsymbol{x}_i = y_{i-1:i-R_l}$ for some $l \leq R_l \ll n$, with the corresponding output being $y_i$. Thus, the dataset $D_n$ can be represented as $(\boldsymbol{x}_i, y_i)_{i=1+R_l}^n$. We apply the same assumptions as for the MLP, specifically 3.1, 3.4, and 3.7. We also use the same definitions and notations for the MLP as those in [20, 21].

Leveraging the mathematical properties of sparse MLPs presented in [20] and the proof of sparse RNNs for a single time series discussed above, one can straightforwardly derive the following Corollary. Let $d(p_1, p_2)$ denote the integrated Hellinger distance between two conditional densities $p_1(y|\boldsymbol{x})$ and $p_2(y|\boldsymbol{x})$. Let $\pi(\cdot|D_n)$ be the posterior probability of an event.

**Corollary B.4.** *Suppose Assumptions 3.1, 3.4, and 3.7 hold. If the mixture Gaussian prior (3) satisfies the conditions : $\lambda_n = O(1/[K_n[n^{H_n}(\bar{L}p_n)]^\tau])$ for some constant $\tau > 0$, $E_n/[H_n \log n + \log \bar{L}]^{1/2} \leq \sigma_{1,n} \leq n^\alpha$ for some constant $\alpha$, and $\sigma_{0,n} \leq \min\{1/[\sqrt{n}K_n(n^{3/2}\sigma_{1,n}/H_n)^{H_n}], 1/[\sqrt{n}K_n(nE_n/H_n)^{H_n}]\}$, then there exists an an error sequence $\epsilon_n^2 = O(\varpi_n^2) + O(\zeta_n^2)$ such that $\lim_{n\to\infty}\epsilon_n = 0$ and $\lim_{n\to\infty} n\epsilon_n^2 = \infty$, and the posterior distribution satisfies*

$$\pi(d(p_{\boldsymbol{\beta}}, p_{\mu^*}) \geq C\epsilon_n|D_n) = O_{P^*}(e^{-n\epsilon_n^2}) \tag{12}$$

*for sufficiently large $n$ and $C > 0$, where $\zeta_n^2 = [r_n H_n \log n + r_n \log \bar{L} + s_n \log p_n]/n$, $p_{\mu^*}$ denotes the underlying true data distribution, and $p_{\boldsymbol{\beta}}$ denotes the data distribution reconstructed by the Bayesian MLP based on its posterior samples.*

# C Asymptotic Normality of Connection Weights and Predictions

This section provides detailed assumptions and proofs for Theorem 3.12 and 3.13. For simplicity, we assume $y_{0:1-R_l}$ is also given, and we let $M_l = R_l + 1$. Let $l_n(\boldsymbol{\beta}) = \frac{1}{n}\sum_{i=1}^n \log(p_{\boldsymbol{\beta}}(y_i|y_{i-1:i-R_l}))$ denote the likelihood function, and let $\pi(\boldsymbol{\beta})$ denote the density of the mixture Gaussian prior (3). Let $h_{i_1,i_2,\ldots,i_d}(\boldsymbol{\beta}) = \frac{\partial^d l_n(\boldsymbol{\beta})}{\partial \beta_{i_1}\cdots\partial \beta_{i_d}}$ which denotes the $d$-th order partial derivatives. Let $H_n(\boldsymbol{\beta})$ denote the Hessian matrix of $l_n(\boldsymbol{\beta})$, and let $h_{i,j}(\boldsymbol{\beta}), h^{i,j}(\boldsymbol{\beta})$ denote the $(i,j)$-th component of $H_n(\boldsymbol{\beta})$ and $H_n^{-1}(\boldsymbol{\beta})$, respectively. Let $B_{\lambda,n} = \bar{\lambda}_n^{1/2}(\boldsymbol{\beta}^*)/\underline{\lambda}_n(\boldsymbol{\beta}^*)$ and $b_{\lambda,n} = \sqrt{r_n/n}B_{\lambda,n}$. For a RNN with $\boldsymbol{\beta}$, we define the weight truncation at the true model structure $\boldsymbol{\gamma}^* : (\boldsymbol{\beta}_{\boldsymbol{\gamma}^*})_i = \boldsymbol{\beta}_i$ for $i \in \boldsymbol{\gamma}^*$ and $(\boldsymbol{\beta}_{\boldsymbol{\gamma}^*})_i = 0$ for $i \notin \boldsymbol{\gamma}^*$. For the mixture Gaussian prior (3), let $B_{\delta_n}(\boldsymbol{\beta}^*) = \{\boldsymbol{\beta} : |\boldsymbol{\beta}_i - \boldsymbol{\beta}_i^*| \leq \delta_n, \forall i \in \boldsymbol{\gamma}^*, |\boldsymbol{\beta}_i - \boldsymbol{\beta}_i^*| \leq 2\sigma_{0,n}\log\left(\frac{\sigma_{1,n}}{\lambda_n\sigma_{0,n}}\right), \forall i \notin \boldsymbol{\gamma}^*\}$.

In addition, we make the following assumptions for Theorem 3.12 and Theorem 3.13.

**Assumption C.1.** Assume the conditions of Lemma 3.10 hold with $\rho(\epsilon_n) = o(\frac{1}{K_n})$ and the $C_1 \geq \frac{2}{3}$ defined in Assumption 3.4. For some $\delta_n$ s.t. $\frac{r_n}{\sqrt{n}} \lesssim \delta_n \lesssim \frac{1}{\sqrt[3]{nr_n}}$, let $A(\epsilon_n, \delta_n) =$

$\{\boldsymbol{\beta} : \max_{i \in \boldsymbol{\gamma}^*} |\boldsymbol{\beta}_i - \boldsymbol{\beta}_i^*| > \delta_n, d(p_{\boldsymbol{\beta}}, p_{\mu^*}) \leq \epsilon_n\}$, where $\epsilon_n$ is the posterior contraction rate as defined in Theorem 3.9. Assume there exists some constants $C > 2$ and $M > 0$ such that

C.1 $\boldsymbol{\beta}^* = (\boldsymbol{\beta}_1, \ldots, \boldsymbol{\beta}_{K_n})$ is generic, $\min_{i \in \boldsymbol{\gamma}^*} |\boldsymbol{\beta}_i^*| > C\delta_n$ and $\pi(A(\epsilon_n, \delta_n)|D_n) \to 0$ as $n \to \infty$.

C.2 $|h_i(\boldsymbol{\beta}^*)| < M, |h_{j,k}(\boldsymbol{\beta}^*)| < M, |h^{j,k}(\boldsymbol{\beta}^*)| < M, |h_{i,j,k}(\boldsymbol{\beta})| < M, |h_l(\boldsymbol{\beta})| < M$ hold for any $i, j, k \in \boldsymbol{\gamma}^*, l \notin \boldsymbol{\gamma}^*$, and $\boldsymbol{\beta} \in B_{2\delta_n}(\boldsymbol{\beta}^*)$.

C.3 $\sup\{|E_{\boldsymbol{\beta}}(a'U^3)| : \|\boldsymbol{\beta}_{\boldsymbol{\gamma}^*} - \boldsymbol{\beta}^*\| \leq 1.2b_{\lambda,n}, \|a\| = 1 \leq 0.1\sqrt{n/r_n}\lambda_n^2(\boldsymbol{\beta}^*)/\bar{\lambda}_n^{1/2}(\boldsymbol{\beta}^*)\}$ and $B_{\lambda,n} = O(1)$, where $U = Z - E_{\boldsymbol{\beta}_{\boldsymbol{\gamma}^*}}(Z)$, $Z$ denotes a random variable drawn from a neural network model parameterized by $\boldsymbol{\beta}_{\boldsymbol{\gamma}^*}$, and $E_{\boldsymbol{\beta}_{\boldsymbol{\gamma}^*}}$ denotes the mean of $Z$.

C.4 $r_n^2/n \to 0$ and the conditions for Theorem 2 of [56] hold.

Conditions $C.1$ to $C.3$ align with the assumptions made in [21]. An additional assumption, $C.4$, is introduced to account for the dependency inherent in time series data. This assumption is crucial and employed in conjunction with $C.3$ to establish the consistency of the maximum likelihood estimator (MLE) of $\boldsymbol{\beta}_{\boldsymbol{\gamma}^*}$ for a given structure $\boldsymbol{\gamma}^*$. While the assumption $r_n/n \to 0$ is sufficient for independent data, which is implied by Assumption 3.4, a stronger restriction, specifically $r_n^2/n \to 0$, is necessary for dependent data such as time series. It is worth noting that the conditions used in Theorem 2 of [56] pertain specifically to the time series data itself.

Let $\mu_{i_1,i_2,\ldots,i_d}(\cdot, \boldsymbol{\beta}) = \dfrac{\partial^d \mu(\cdot, \boldsymbol{\beta})}{\partial \beta_{i_1} \cdots \partial \beta_{i_d}}$ denotes the d-th order partial derivative for some input.

**Assumption C.2.** $|\mu_i(\cdot, \boldsymbol{\beta}^*)| < M, |\mu_{i,j}(\cdot, \boldsymbol{\beta})| < M, |\mu_k(\cdot, \boldsymbol{\beta})| < M$ for any $i, j \in \boldsymbol{\gamma}^*, k \notin \boldsymbol{\gamma}^*$, and $\boldsymbol{\beta} \in B_{2\delta_n}(\boldsymbol{\beta}^*)$, where $M$ is as defined in Assumption C.1.

**Proof of Theorem 3.12 and Theorem 3.13**  The proof of these two theorems can be conducted following the approach in [21]. The main difference is that in [21], they rely on Theorem 2.1 of [57], which assumes independent data. However, the same conclusions can be extended to time series data by using Theorem 2 of [56], taking into account assumption $C.4$. This allows for the consideration of dependent data in the analysis.

## D  Computation

Algorithm 1 gives the prior annealing procedure[21]. In practice, the following implementation can be followed based on Algorithm 1:

- For $0 < t < T_1$, perform initial training.
- For $T_1 \leq t \leq T_2$, set $\sigma_{0,n}^{(t)} = \sigma_{0,n}^{init}$ and gradually increase $\eta^{(t)} = \frac{t-T_1}{T_2-T_1}$.
- For $T_2 \leq t \leq T_3$, set $\eta^{(t)} = 1$ and gradually decrease $\sigma_{0,n}^{(t)}$ according to the formula: $\sigma_{0,n}^{(t)} = \left(\frac{T_3-t}{T_3-T_2}\right)(\sigma_{0,n}^{init})^2 + \left(\frac{t-T_2}{T_3-T_2}\right)(\sigma_{0,n}^{end})^2$.
- For $t > T_3$, set $\eta^{(t)} = 1$, $\sigma_{0,n}^{(t)} = \sigma_{0,n}^{end}$, and gradually decrease the temperature $\tau$ according to the formula: $\tau = \frac{c}{t-T_3}$, where $c$ is a constant.

Please refer to Appendix E for intuitive explanations of the prior annealing algorithm and further details on training a model to achieve the desired sparsity.

## E  Mixture Gaussian Prior

The mixture Gaussian prior imposes a penalty on the model parameters by acting as a piecewise L2 penalty, applying varying degrees of penalty in different regions of the parameter space. Given values for $\sigma_{0,n}$, $\sigma_{1,n}$, and $\lambda_n$, a threshold value can be computed using Algorithm

---

**Algorithm 1** Prior Annealing: Frequentist

---

[1] (Initial Training) Train a neural network satisfying condition $(S^*)$ such that a global optimal solution $\boldsymbol{\beta}_0 = \arg\max_{\boldsymbol{\beta}} l_n(\boldsymbol{\beta})$ is reached, this stage can be accomplished by using SGD or Adam.

[2] (Prior Annealing) Initialize $\boldsymbol{\beta}$ at $\boldsymbol{\beta}_0$, and simulate from a sequence of distributions $\pi(\boldsymbol{\beta}|D_n, \tau, \eta^{(k)}, \sigma_{0,n}^{(k)}) \propto e^{nl_n(\boldsymbol{\beta})/\tau} \pi_k^{\eta^{(k)}/\tau}(\boldsymbol{\beta})$ for $k = 1, 2, \ldots, m$, where $0 < \eta^{(1)} \leq \eta^{(2)} \leq \cdots \leq \eta^{(m)} = 1$, $\pi_k = \lambda_n N(0, \sigma_{1,n}^2) + (1 - \lambda_n)N(0, (\sigma_{0,n}^{(k)})^2)$, and $\sigma_{0,n}^{init} = \sigma_{0,n}^{(1)} \geq \sigma_{0,n}^{(2)} \geq \cdots \geq \sigma_{0,n}^{(m)} = \sigma_{0,n}^{end}$. This can be done by using stochastic gradient MCMC algorithms [58]. After the stage $m$ has been reached, continue to run the simulated annealing algorithm by gradually decreasing the temperature $\tau$ to a very small value. Denote the resulting model by $\hat{\boldsymbol{\beta}}$.

[3] (Structure Sparsification) For each connection weight $i \in \{1, 2, \ldots, K_n\}$, set $\tilde{\boldsymbol{\gamma}}_i = 1$, if $|\boldsymbol{\beta}_i| > \dfrac{\sqrt{2}\sigma_{0,n}\sigma_{1,n}}{\sqrt{\sigma_{1,n}^2 - \sigma_{0,n}^2}}\sqrt{\log\left(\dfrac{1 - \lambda_n}{\lambda_n}\dfrac{\sigma_{1,n}}{\sigma_{0,n}}\right)}$ and 0 otherwise, where the threshold value is determined by solving $\pi(\boldsymbol{\gamma}_i = 1|\boldsymbol{\beta}) > 0.5$, and $\sigma_{0,n} = \sigma_{0,n}^{end}$. Denote the the structure of the sparse RNN by $\tilde{\boldsymbol{\gamma}}$.

[4] (Nonzero-weights Refining) Refine the nonzero weights of the sparse model $(\hat{\boldsymbol{\beta}}, \tilde{\boldsymbol{\gamma}})$ by maximizing $l_n(\boldsymbol{\beta})$. Denote the resulting estimate by $(\tilde{\boldsymbol{\beta}}, \tilde{\boldsymbol{\gamma}})$.

---

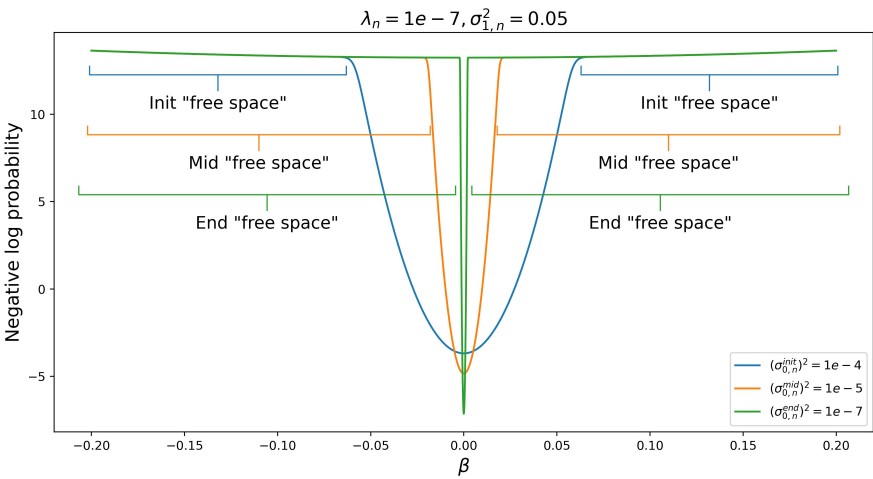

Figure 3: Changes of the negative log prior density function in prior annealing of $\sigma_{0,n}^2$, where the values of $\lambda_n$ and $\sigma_{1,n}^2$ are fixed.

1. Parameters whose absolute values are below this threshold will receive a large penalty, hence constituting the "penalty region", while parameters whose absolute values are above the threshold will receive relatively minimal penalty, forming the "free space". The severity of the penalty in the free space largely depends on the value of $\sigma_{1,n}$.

For instance, as depicted in Figure 3, based on the simple practice implementation detailed in Appendix D, we fix $\lambda_n = 1e - 7$, $\sigma_{1,n}^2 = 0.05$, and we set $(\sigma_{0,n}^{init})^2 = 1e - 5$, $(\sigma_{0,n}^{end})^2 = 1e - 7$, and gradually reduce $\sigma_{0,n}^2$ from the initial value to the end value. Initially, the free space is relatively small and the penalty region is relatively large. However, the penalty imposed on the parameters within the initial penalty region is also minor, making it challenging to shrink these parameters to zero. As we progressively decrease $\sigma_{0,n}^2$, the free space enlarges,

the penalty region diminishes, and the penalty on the parameters within the penalty region intensifies simultaneously. Once $\sigma_{0,n}^2$ equals $(\sigma_{0,n}^{end})^2$, the parameters within the penalty region will be proximate to zero, and the parameters outside the penalty region can vary freely in almost all areas of the parameter space.

For model compression tasks, achieving the desired sparsity involves several steps post the initial training phase outlined in Algorithm 1. First, determine the initial threshold value based on pre-set values for $\lambda_n$, $\sigma_{1,n}^2$, and $(\sigma_{0,n}^{init})^2$. Then, compute the proportion of the parameters in the initial model whose absolute values are lower than this threshold. This value serves as an estimate for anticipated sparsity. Adjust the $(\sigma_{0,n}^{init})^2$ value until the predicted sparsity aligns closely with the desired sparsity level.

## F  Construction of Multi-Horizon Joint Prediction Intervals

To construct valid joint prediction intervals for multi-horizon forecasting, we estimate the individual variances for each time step in the prediction horizon using procedures similar to one-step-ahead forecasting. We then adjust the critical value by dividing $\alpha$ by the number of time steps $m$ in the prediction horizon and use $z_{\alpha/(2m)}$ as the adjusted critical value. This Bonferroni correction ensures the desired coverage probability across the entire prediction horizon.

As an example, we refer to the experiments conducted in 5.1.2, where we work with a set of training sequences denoted by $D_n = \{y_1^{(i)}, \ldots, y_T^{(i)}, \ldots y_{T+m}^{(i)}\}_{i=1}^n$. Here, $y_{T+1}^{(i)}, y_{T+2}^{(i)}, \ldots, y_{T+m}^{(i)}$ represent the prediction horizon for each sequence, while $T$ represents the length of the observed sequence.

- Train a model by the proposed algorithm, and denote the trained model by $(\hat{\boldsymbol{\beta}}, \hat{\boldsymbol{\gamma}})$.
- Calculate $\hat{\boldsymbol{\sigma}}^2$ as an estimator of $\boldsymbol{\sigma}^2 \in \mathbb{R}^{m \times 1}$:

$$\hat{\boldsymbol{\sigma}}^2 = \frac{1}{n-1} \sum_{i=1}^n \left( \boldsymbol{\mu}(\hat{\boldsymbol{\beta}}, y_{1:T}^{(i)}) - y_{T+1:T+m}^{(i)} \right) \otimes \left( \boldsymbol{\mu}(\hat{\boldsymbol{\beta}}, y_{1:T}^{(i)}) - y_{T+1:T+m}^{(i)} \right),$$

  where $\boldsymbol{\mu}(\hat{\boldsymbol{\beta}}, y_{1:T}^{(i)}) = \hat{y}_{T+1:T+m}^{(i)}$, and $\otimes$ denotes elementwise product.
- For a test sequence $y_{1:T}^{(0)}$, calculate

$$\hat{\boldsymbol{\Sigma}} = \nabla_{\hat{\boldsymbol{\gamma}}} \boldsymbol{\mu}(\hat{\boldsymbol{\beta}}, y_{1:T}^{(0)})' (-\nabla_{\hat{\boldsymbol{\gamma}}}^2 l_n(\hat{\boldsymbol{\beta}}))^{-1} \nabla_{\hat{\boldsymbol{\gamma}}} \boldsymbol{\mu}(\hat{\boldsymbol{\beta}}, y_{1:T}^{(0)}).$$

  Let $\hat{\boldsymbol{\varsigma}} \in \mathbb{R}^{m \times 1}$ denote the vector formed by the diagonal elements of $\hat{\boldsymbol{\Sigma}}$.
- The Bonferroni simultaneous prediction intervals for all elements of $y_{T+1:T+m}^{(0)}$ are given by

$$\boldsymbol{\mu}(\hat{\boldsymbol{\beta}}, y_{1:T}^{(0)}) \pm z_{\alpha/(2m)} \left( \frac{1}{n} \hat{\boldsymbol{\varsigma}} + \hat{\boldsymbol{\sigma}}^2 \right)^{\circ \frac{1}{2}},$$

  where $\circ \frac{1}{2}$ represents the element-wise square root operation.

The Bayesian method is particularly advantageous when dealing with a large number of non-zero connection weights in $(\hat{\boldsymbol{\beta}}, \hat{\boldsymbol{\gamma}})$, making the computation of the Hessian matrix of the log-likelihood function costly or unfeasible. For detailed information on utilizing the Bayesian method for constructing prediction intervals, please refer to [21].

## G  Numerical Experiments

### G.1  French Electricity Spot Prices

**Dataset.** The given dataset contains the spot prices of electricity in France that were established over a period of four years, from 2016 to 2020, using an auction market. In

this market, producers and suppliers submit their orders for the next day's 24-hour period, specifying the electricity volume in MWh they intend to sell or purchase, along with the corresponding price in €/MWh. At midnight, the Euphemia algorithm, as described in [59], calculates the 24 hourly prices for the next day, based on the submitted orders and other constraints. This hourly dataset consists of 35064 observations, covering $(3 \times 365 + 366) \times 24$ periods. Our main objective is to predict the 24 prices for the next day by considering different explanatory variables such as the day-ahead forecast consumption, day-of-the-week, as well as the prices of the previous day and the same day in the previous week, as these variables are crucial in determining the spot prices of electricity. Refer to Figure 4 for a visual representation of the dataset.

The prediction models and training settings described below are the same for all 24 hours.

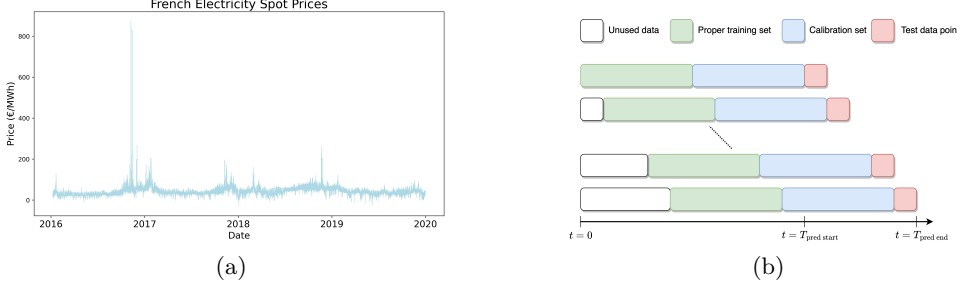

(a)                   (b)

Figure 4: (a): French electricity spot prices (2016 to 2019). (b): Online learning method used for ACI and AgACI in [31], where the prediction range is assumed to between $T_{\text{pred start}}$ and $T_{\text{pred end}}$.

**Prediction Model.** We use an MLP with one hidden layer of size 100 and the sigmoid activation function as the underlying prediction model for all methods and all hours.

**Training: Baselines.** For all CP baselines, we train the MLP for 300 epochs using SGD with a constant learning rate of 0.001 and momentum of 0.9, the batch size is set to be 100. For the ACI, we use the same list of $\gamma \in \{0, 0.000005, 0.00005, 0.0001, 0.0002, 0.0003, 0.0004, 0.0005, 0.0001, 0.0002, 0.0003, 0.0004, 0.0005, 0.0006, 0.0007, 0.0008, 0.0009, 0.001, 0.002, 0.003, 0.004, 0.005, 0.006, 0.007, 0.008, 0.009, 0.01, 0.02, 0.03, 0.04, 0.05, 0.06, 0.07, 0.08, 0.09\}$ as in [31].

**Training: Our Method.** For our method PA, we train a total of 300 epochs with the same learning rate, momentum, and batch size. We use $T_1 = 150$, $T_2 = 160$, $T_3 = 260$. We run SGD with momentum for $t < T_1$ and SGHMC with temperature= 1 for $t >= T_1$. For the mixture Gaussian prior, we fix $\sigma_{1,n}^2 = 0.01$, $(\sigma_{1,n}^{init})^2 = 1e-5$, $(\sigma_{1,n}^{end})^2 = 1e-6$, and $\lambda_n = 1e-7$.

Table 5: Uncertainty quantification results produced by different methods for the French electricity spot prices dataset. This table corresponds to the Figure 2.

| Methods | Coverage | Average PI Lengths (standard deviation) | Median PI Lengths (interquartile range) |
|---|---|---|---|
| PA offline (**ours**) | 90.06 | 20.19(3.23) | 20.59(1.82) |
| AgACI online | 91.05 | 23.27(14.49) | 21.38(2.62) |
| ACI $\gamma = 0.01$ online | 90.21 | 22.13(8.50) | 20.68(2.70) |
| ACI $\gamma = 0.0$ online | 92.89 | 24.39(9.17) | 22.99(2.91) |
| EnbPI V2 online | 91.23 | 27.99(10.17) | 26.55(5.09) |
| NexCP online | 91.18 | 24.41(10.40) | 22.88(2.86) |

## G.2   EEG, MIMIC-III, and COVID-19

**EEG** The EEG dataset, available at here, served as the primary source for the EEG signal time series. This dataset contains responses from both control and alcoholic subjects who were presented with visual stimuli of three different types. To maintain consistency with

previous work [33], we utilized the medium-sized version, consisting of 10 control and 10 alcoholic subjects. We focused solely on the control subjects for our experiments, as the dataset summaries indicated their EEG responses were more difficult to predict. For detailed information on our processing steps, please refer to [33].

**COVID-19** We followed the processing steps of previous research [33] and utilized COVID-19 data from various regions within the same country to minimize potential distribution shifts while adhering to the exchangeability assumption. The lower tier local authority split provided us with 380 sequences, which we randomly allocated between the training, calibration, and test sets over multiple trials. The dataset can be found at here.

**MIMIC-III** We collect patient data on the use of antibiotics (specifically Levofloxacin) from the MIMIC-III dataset [46]. However, the subset of the MIMIC-III dataset used in [33] is not publicly available to us, and the authors did not provide information on the processing steps, such as SQL queries and Python code. Therefore, we follow the published processing steps provided in [60]. We remove sequences with fewer than 5 visits or more than 47 visits, resulting in a total of 2992 sequences. We randomly split the dataset into train, calibration, and test sets, with corresponding proportions of 43%, 47%, and 10%. We use the white blood cell count (high) as the feature for the univariate time series from these sequences. Access to the MIMIC-III dataset requires PhysioNet credentialing.

Table 6 presents the training hyperparameters and prediction models used for the baselines in the three datasets, while Table 7 presents the training details used for our method.

We hypothesize that the mediocre performance on the EEG dataset is due to the small size of the prediction model, which has only 2025 parameters and is substantially smaller than the number of training sequences. Hence, we conducted experiments on the EEG dataset using a slightly larger prediction model for different methods. The results are presented in Table 8, and the corresponding training details are provided in Table 6. Again, our method outperforms other baselines as well.

Table 6: Training details for the baselines, where LSTM-$d$ refers to a single hidden layer LSTM network with a size of $d$.

|  | **MIMIC-III** | **EEG** | **COVID-19** |
|---|---|---|---|
| model | LSTM-500 | LSTM-20 | LSTM-20 |
| learning rate | $2e-4$ | 0.01 | 0.01 |
| Epochs | 65 | 100 | 1000 |
| Optimizer | Adam | Adam | Adam |
| Batch size | 150 | 150 | 150 |

Table 7: Training details for our method, where LSTM-$d$ refers to a single hidden layer LSTM network with a size of $d$.

|  | **MIMIC-III** | **EEG** | **COVID-19** |
|---|---|---|---|
| Model | LSTM-500 | LSTM-20 | LSTM-20 |
| Learning rate | $2e-4$ | 0.01 | 0.01 |
| Epochs | 65 | 100 | 1000 |
| Optimizer | Adam | Adam | Adam |
| Batch size | 150 | 150 | 150 |
| $\sigma_{1,n}^2$ | 0.05 | 0.1 | 0.1 |
| $(\sigma_{0,n}^{init})^2$ | $1e-5$ | $1e-7$ | $1e-7$ |
| $(\sigma_{0,n}^{end})^2$ | $1e-6$ | $1e-8$ | $1e-8$ |
| $\lambda_n$ | $1e-7$ | $1e-7$ | $1e-7$ |
| Temperature | 1 | 1 | 1 |
| $T_1$ | 40 | 50 | 800 |
| $T_2$ | 40 | 55 | 850 |
| $T_3$ | 55 | 80 | 950 |

Table 8: Uncertainty quantification results produced by various methods for the EEG data using a larger prediction model. The "coverage" represents the average joint coverage rate over different random seeds. The prediction interval (PI) lengths are averaged across prediction horizons and random seeds. The values in parentheses indicate the standard deviations of the respective means.

| | EEG | |
|---|---|---|
| **Model** | Coverage | PI lengths |
| PA-RNN | 91.0%(0.8%) | 40.84(5.69) |
| CF-RNN | 90.8%(1.8%) | 43.4(6.79) |
| MQ-RNN | 45.2%(2.5%) | 20.63(2.07) |
| DP-RNN | 0.6%(0.1%) | 5.02(0.53) |

Table 9: Training details for the EEG data with a slightly larger prediction model for all methods, where "-" denotes that the information is not applicable.

| | PA-RNN | CF-RNN | MQ-RNN | DP-RNN |
|---|---|---|---|---|
| model | LSTM-100 | LSTM-100 | LSTM-100 | LSTM-100 |
| learning rate | $1e-3$ | $1e-3$ | $1e-3$ | $1e-3$ |
| Epochs | 150 | 150 | 150 | 150 |
| Optimizer | Adam | Adam | Adam | Adam |
| Batch size | 150 | 150 | 150 | 150 |
| $\sigma_{1,n}^2$ | 0.01 | - | - | - |
| $(\sigma_{0,n}^{init})^2$ | $1e-5$ | - | - | - |
| $(\sigma_{0,n}^{end})^2$ | $1e-6$ | - | - | - |
| $\lambda_n$ | $1e-7$ | - | - | - |
| temperature | 1 | - | - | - |
| $T_1$ | 100 | - | - | - |
| $T_2$ | 105 | - | - | - |
| $T_3$ | 130 | - | - | - |

## G.3 Autoregressive Order Selection

**Model** An Elman RNN with one hidden layer of size 1000. Different window sizes (i.e., 1 or 15) will result in a different total number of parameters..

**Hyperparameters** All training hyperparameters are given in Table 10

Table 10: Training details for the autoregressive order selection experiment.

| | PA-RNN 1 | PA-RNN 15 | RNN 1 | RNN 15 |
|---|---|---|---|---|
| Learning rate | $4e-3$ | $1e-4$ | $1e-4$ | $1e-4$ |
| Iterations | 25000 | 25000 | 25000 | 25000 |
| Optimizer | SGHMC | SGHMC | SGHMC | SGHMC |
| Batch size | 36 | 36 | 36 | 36 |
| Subsample size per iteration | 50 | 50 | 50 | 50 |
| Predicton horizon | 1 | 1 | 1 | 1 |
| $\sigma_{1,n}^2$ | 0.05 | 0.05 | - | - |
| $(\sigma_{0,n}^{init})^2$ | $2e-6$ | $4e-6$ | - | - |
| $(\sigma_{0,n}^{end})^2$ | $1e-7$ | $1e-7$ | - | - |
| $\lambda_n$ | $1e-7$ | $1e-7$ | - | - |
| Temperature | 0.1 | 0.1 | - | - |
| $T_1$ | 5000 | 5000 | - | - |
| $T_2$ | 10000 | 10000 | - | - |
| $T_3$ | 25000 | 25000 | - | - |

## G.4    Large-Scale Model Compression

As pointed out by recent summary/survey papers on sparse deep learning [61, 62], the lack of standardized benchmarks and metrics that provide guidance on model structure, task, dataset, and sparsity levels has caused difficulties in conducting fair and meaningful comparisons with previous works. For example, the task of compressing Penn Tree Bank (PTB) word language model [63] is a popular comparison task. However, many previous works [64, 65, 66, 67] have avoided comparison with the state-of-the-art method by either not using the standard baseline model, not reporting, or conducting comparisons at different sparsity levels. Therefore, we performed an extensive search of papers that reported performance on this task, and to the best of our knowledge, the state-of-the-art method is the Automated Gradual Pruning (`AGP`) by [68].

In our experiments, we train large stacked LSTM language models on the PTB dataset at different sparsity levels. The model architecture follows the same design as in [63], comprising an embedding layer, two stacked LSTM layers, and a softmax layer. The vocabulary size for the model is 10000, the embedding layer size is 1500, and the hidden layer size is 1500 resulting in a total of 66 million parameters. We compare our method with `AGP` at different sparsity levels, including 80%, 85%, 90%, 95%, and 97.5% as in [68]. The results are summarized in Table 11, and our method achieves better results consistently. For the `AGP`, numbers are taken directly from the original paper, and since they only provided one experimental result for each sparsity level, no standard deviation is reported. For our method, we run three independent trials and provide both the mean and standard deviation for each sparsity level. During the initial training stage, we follow the same training procedure as in [63]. The details of the prior annealing and fine-tuning stages of our method for different sparsity levels are provided below.

During the initial training stage, we follow the same training procedure as in [63] for all sparsity, and hence $T_1 = 55$.

During the prior annealing stage, we train a total of 60 epochs using SGHMC. For all levels of sparsity considered, we fix $\sigma_{1,n}^2 = 0.5$, $\lambda_n = 1e - 6$, momentum $1 - \alpha = 0.9$, minibatch size $= 20$, and we fix $T_2 = 5 + T_1 = 60, T_3 = T_2 + 20 = 80$. We set the initial temperature $\tau = 0.01$ for $t \le T_3$ and and gradually decrease $\tau$ by $\tau = \dfrac{0.01}{t - T_3}$ for $t > T_3$.

During the fine tune stage, we apply a similar training procedure as the initial training stage, i.e., we use SGD with gradient clipping and we decrease the learning rate by a constant factor after certain epochs. The minibatch size is set to be 20, and we apply early stopping based on the validation perplexity with a maximum of 30 epochs.

Table 12 gives all hyperparameters (not specified above) for different sparsity levels.

Note that, the mixture Gaussian prior by nature is a regularization method, so we lower the dropout ratio during the prior annealing and fine tune stage for models with relatively high sparsity.

Table 11: Test set perplexity for different methods and sparsity levels.

| Method | Sparsity | Test Perplexity |
|--------|----------|-----------------|
| baseline | 0% | 78.40 |
| PA | $80.87\% \pm 0.09\%$ | $77.122 \pm 0.147$ |
| PA | $85.83\% \pm 0.05\%$ | $77.431 \pm 0.109$ |
| PA | $90.53\% \pm 0.05\%$ | $78.823 \pm 0.118$ |
| PA | $95.40\% \pm 0.00\%$ | $84.525 \pm 0.112$ |
| PA | $97.78\% \pm 0.05\%$ | $93.268 \pm 0.136$ |
| AGP | 80% | 77.52 |
| AGP | 85% | 78.31 |
| AGP | 90% | 80.24 |
| AGP | 95% | 87.83 |
| AGP | 97.5% | 103.20 |

Table 12: Word Language Model Compression: Hyperparameters for our method during the prior annealing and fine tune stage. We denote the learning rate by LR, the prior annealing stage by PA, and the fine tune stage by FT.

| Hyperparameters/Sparsity | 80% | 85% | 90% | 95% | 97.5% |
|--------------------------|-----|-----|-----|-----|-------|
| Dropout ratio | 0.65 | 0.65 | 0.65 | 0.4 | 0.4 |
| $(\sigma_{0,n}^{init})^2$ | 0.0005 | 0.0007 | 0.00097 | 0.00174 | 0.00248 |
| $(\sigma_{0,n}^{end})^2$ | $1e-7$ | $1e-7$ | $1e-7$ | $1e-7$ | $1e-6$ |
| LR PA | 0.004 | 0.004 | 0.008 | 0.008 | 0.01 |
| LR FT | 0.008 | 0.008 | 0.008 | 0.1 | 0.2 |
| LR decay factor FT | 1/0.95 | 1/0.95 | 1/0.95 | 1/0.95 | 1/0.95 |
| LR decay epoch FT | 5 | 5 | 5 | 5 | 5 |

Table 13: Training details for the additional autoregressive order selection experiment.

| $y_i = \left(0.8 - 1.1\exp\left\{-50y_{i-1}^2\right\}\right)y_{i-1} + \eta_i$ | PA-RNN $1, 3, 5, 7, 10, 15$ |
|---|---|
| Learning rate | $1e-4$ |
| Iterations | 2000 |
| Optimizer | SGHMC |
| Batch size | 36 |
| Subsample size per iteration | 50 |
| Predicton horizon | 1 |
| $\sigma_{1,n}^2$ | 0.05 |
| $(\sigma_{0,n}^{init})^2$ | $1e-5$ |
| $(\sigma_{0,n}^{end})^2$ | $1e-6$ |
| $\lambda_n$ | $1e-6$ |
| Temperature | 0.1 |
| $T_1$ | 500 |
| $T_2$ | 1000 |
| $T_3$ | 1500 |

