# OpenReview forum: "Sparse Deep Learning for Time Series Data: Theory and Applications"
_NeurIPS.cc/2023/Conference — NeurIPS 2023 poster_

### Official Review · Reviewer_shpG · 2023-07-02

**Soundness:** 3 good
**Presentation:** 3 good
**Contribution:** 3 good
**Rating:** 7
**Confidence:** 3

**Summary:**

This paper introduces an expansion of sparse deep learning theory, specifically tailored for the analysis of time series data. The authors provide a detailed explanation of the theoretical foundation behind this theory, employing recurrent neural networks (RNNs), particularly long short-term memory (LSTM) networks. The authors demonstrate that their novel approach yields competitive outcomes when compared to other cutting-edge methodologies from diverse research domains, including conformal prediction.

**Strengths:**

The paper is well-written and grammatically correct with no evident typos. The narrative is clear: related work, theoretical grounding, computation and experiments.

It is clear that the authors have well investigated the theory of sparse deep learning and it is evident that they know it quite well the field. Hence, the derivations, in principle, seem correct — although I must admit that I found the ideas and derivations presented in the paper to be somewhat challenging to grasp, even though I devoted a considerable amount of time to understanding them. Plus, newcomers to the sparse deep learning theory can find the complete derivations in the Appendix, which is quite extensive.

The proposed method is tested in three relevant tasks related to time series data, and it is evaluated on at least four real-world datasets. This thorough evaluation approach provides strong empirical evidence to support the theoretical foundations of the method, demonstrating its validity and applicability.

**Weaknesses:**

Before outlining the weaknesses, I repeat again that I am a new reader of sparse deep learning theory. However, I still find that the paper might exhibit some weaknesses:

1. First of all, while I believe the theoretical side of the paper is of good quality, I encountered difficulties in understanding the concepts due to the overwhelming presence of complex notation. Sometimes there are many variables, and many subscripts, which in temporal series can be, e.g. $y_{k-1:k-R_l}$. This makes the reading difficult and sometimes cumbersome.
2. After reading the paper, I still cannot see why do you choose conformal prediction as the main comparison field. I lack some more context or motivation on the paper. The paper might be already interesting for the natural extension of sparse deep learning for time series data, but I also believe there should be some kind of justification of why these methods are a good alternative.
3. The point above leads me to the following point: I worry that the literature review for Conformal Prediction might be scarce. Authors comment on the work by [1], but there is no reference to other works in the same line of this work [3], or other contemporary works such as [2].
4. And following this line, I wonder if this theory might be very challenging to apply in real-world scenarios compared to CP. The beauty of CP is that you can basically apply it on top of any ML model in a “plug-and-play” fashion, with the cost of very strong assumptions such as the ****************exchangeability**************** assumption. But it is more straightforward to apply and to easily understand than this sparse deep learning technique. I would like to know what the authors think about this.

[1] Isaac Gibbs and Emmanuel Candes. Adaptive conformal inference under distribution shift. Advances in Neural Information Processing Systems, 34:1660–1672, 2021.

[2] Barber, R. F., Candes, E. J., Ramdas, A., & Tibshirani, R. J. (2023). Conformal prediction beyond exchangeability. *The Annals of Statistics*, *51*(2), 816-845.

[3] Romano, Y., Patterson, E., & Candes, E. (2019). Conformalized quantile regression. *Advances in neural information processing systems*, *32*.

**Questions:**

Here are some questions/suggestions for the authors:

- In Figure 1, the circles shouldn’t contain some variables or some naming? Are these the states of the RNN? This figure is a bit misleading to me.
- In line 239, what is ******local-trap issue?:****** A bit more context could help the reader.
- Is there any reason why you choose LSTM against other architectures? Some study on whether this applies to GRU or other RNN could be interesting.
- Do you believe that the LSTM is providing good results because of the forget gate? Or do you think there is some alignment between your results and the use of LSTM, that might actually shadow the developed theroy?
- In Table 3, when considering the autoregressive order selection, it appears that the errors of the proposed model are comparable to those of the baselines that only employ RNN. Notably, the absence of standard errors in the results raises some questions. It is possible that I may have overlooked something, or perhaps the authors could provide additional clarity regarding the significance of these findings. One noteworthy aspect is whether the reduced number of hidden links is the primary factor contributing to the performance of the proposed method.

Minor questions:

- What is PI length? Prediction Interval length?  It is just presented in the table with no extra explanation.
- Why the method is called PA? When do you choose the name?

**Limitations:**

The authors did not explicitly address the limitations of their work, which raises concerns about the feasibility of deploying their proposed model in practical settings. It is crucial to understand the potential challenges and constraints associated with implementation. Considering the computational time and power required, it is essential to evaluate the trade-off involved and compare it to the relatively inexpensive nature of conformal prediction. Understanding these aspects would provide valuable insights into the practical implications and potential drawbacks of the proposed model. Plus, I lack some direct comments on the limitations with respect to conformal prediction, since my prior belief is that the their presented comparison w.r.t conformal prediction methods is scarce.

With all this in mind, I consider that the paper needs some further improvement, and I await for the author’s and other reviewers’ comments for the possibility of increasing the score.

---

> ### Author Rebuttal · Authors · 2023-08-09
>
> Thank you so much for your review. We provide a point-by-point response to your comments below.
>
> $\textbf{W1: Complex notation}$
>
> In the revision, we will simplify the notation and add more explanations to the assumptions and results, making the paper more accessible to the readers.
>
> $\textbf{W2: Conformal prediction baselines}$
>
> We chose conformal prediction as the main comparison field because conformal prediction methods have gained significant popularity and adoption as a widely-used approach for uncertainty quantification in various domains, including time series data. By selecting CP as the main comparison field, our aim is to assess the performance of our proposed method against established and widely-recognized approaches for uncertainty estimation.
>
> In our experiments, we compared the proposed method with two state-of-the-art CP methods for time series data [1,4]. As suggested, we have included another recently published conformal baseline focused on time series (EnbPI V2) [5] for comparison in the single time series experiment (please see table below). We greatly appreciate the reviewer's input on the literature review. In the revised manuscript, we will provide a more comprehensive review, including references to [2,3].
>
> It's worth noting that [1] presents an adaptive version of [3] specifically designed for data with potential distribution shifts, such as time series data. Additionally, [2] introduced a random swapping mechanism to address potentially non-exchangeable data. Their extension enables conformal prediction to be applied on top of a model trained with weighted samples. Their primary focus is providing theoretical justification of the gap in the coverage rate for the proposed method.
>
> |   Methods   | Coverage | Average PI Lengths (standard deviation)  | Median PI Lengths (interquartile range) |
> | -------- | -------- | -------- | -------- |
> | PA offline | 90.06 | 20.19(3.23)  | 20.59(1.82) |
> | AgACI online  | 91.05 | 23.27(14.49)  | 21.38(2.62) |
> | ACI $\gamma=0.01$  online | 90.21  | 22.13(8.50)  | 20.68(2.70) |
> | ACI $\gamma=0$ online | 92.89  | 24.39(9.17)  | 22.99(2.91) |
> | EnbPI V2 online [5] | 91.23  | 27.99(10.17)  | 26.55(5.09)|
>
> $\textbf{W3: Our method applied in real-world scenarios compared to CP methods}$
>
> We appreciate the reviewer's comments . While basic CP methods like split conformal [6] are easy to apply on top of any ML models to generate prediction sets with marginal coverage guarantee, the width or the usefulness of the prediction sets will depend on how well the ML model performs or how the non-conformity scores are defined. Our theoretical results provide convergence to the true model, so ML models will perform well under our assumptions. Moreover, advanced CP methods like [4] and [5] can require multiple runs and ensemble models, making them computationally intensive, especially for deep neural networks. In contrast, our sparse deep learning technique offers several advantages in real-world scenarios. Once the model is trained with the mixture Gaussian priors, which act as an alternative to L1 or L2 regularization, the prediction intervals become a byproduct of the training process. This means that no extra training or heavy computation is needed to produce prediction intervals.
>
> $\textbf{Q1: Figure 1}$
>
> We apologize for the confusion. The circles in Figure 1 represent the hidden states of a multi-layer RNN. To avoid ambiguity, we will include explicit labeling of the variables in the camera-ready version to enhance clarity.
>
> $\textbf{Q2: What is local-trap issue?}$
>
> Apologies for the confusion. In this context, the term 'local-trap' refers to the same phenomenon as 'local minima.' There has been substantial research examining the favorable properties of the optimization landscape for over-parameterized neural networks, e.g., [7], which aims to explain why gradient-based methods work well for highly non-convex neural network optimization problems. However, for our model, the mixture Gaussian prior introduces heavy penalties on parameters near $0$, so it is unclear how the optimization landscape would be affected and local traps could become an issue. That is the motivation behind applying the prior annealing approach.
>
> $\textbf{Q3: Is there any reason why you choose LSTM?}$
>
> We chose LSTM as the main network in our experimental setup to facilitate fair comparisons with current state-of-the-art methods for tasks like time series forecasting and NLP. Many of these existing methods employ LSTM, and by using the same network architecture, we can better assess the effectiveness of our proposed method relative to these benchmarks.
>
> $\textbf{Q4: Table 3}$
>
> Thank you for your feedback. In our table, we reported the corresponding standard deviations for all metrics except FSR and NSR. The reason for the absence of standard deviations for FSR and NSR is that their calculation requires variable selection results from all five datasets, making it infeasible to compute standard deviations for these metrics. If you require further clarification on any aspect, please feel free to let us know.
>
> $\textbf{Q5 and Q6: PI and PA:}$
>
> Yes, you are correct. "PI length" stands for Prediction Interval length. To clarify this in the revised version, we will add a brief explanation of "PI length" in the table caption. We chose the name 'PA' (Prior Annealing) for our method because a crucial aspect of our algorithm involves annealing the mixture Gaussian prior.
>
> [1] Adaptive conformal inference under distribution shift. NeurIPS2021.
>
> [2] Conformal prediction beyond exchangeability. The Annals of Statistics, 2023.
>
> [3] Conformalized quantile regression. NeurIPS2019.
>
> [4] Adaptive conformal predictions for time series. ICML2022.
>
> [5] Conformal prediction interval for dynamic time. ICML2021.
>
> [6] Distribution-free predictive inference for regression. JASA2018.
>
> [7] The loss surface of deep and wide. ICML2017.

---

> > ### Comment · Reviewer_shpG · 2023-08-15
> > **Update score**
> >
> > I thank the authors for the detailed response. I believe that most of my concerns were properly addressed, and I checked that I shared some concerns with other reviewers who also worried/struggled to grasp the idea of the paper at certain parts because of the complex notation.
> >
> > ### W2: Conformal Prediction Baselines
> > The authors correctly addressed my concerns and investigated other conformal methods for time series data. However, is there any reason why the authors omitted the comparison w.r.t [1]? I think this work would greatly improve with this comparison and would become more relevant for other researchers using conformal prediction since this work is a reference work for CP under no exchangeability assumption.
> >
> > ### W3: Comparison to CP methods in real-world datasets
> > I have still doubts about how sound the method can be in real-world scenarios. I mean, conformal prediction becomes popular because it can be __easily__ adapted in any ML model. However, the proposed method by the authors must be tuned for so many hyperparameters, there is lots of theory that must be digested to fully understand the method, etc. So I believe it cannot be an easy tool to deploy in practice. I believe the authors properly answered this concern, but maybe some direct comments about the limitations of the method in practice: training time, hyperparameter sensibility for performance, etc.?
> >
> > Regarding the questions, I believe the authors properly addressed my concerns.
> >
> > Overall, I think the authors answered my concerns, as well as other reviewers' concerns. I update my score accordingly, and I await the discussion with the other reviewers for the final decision. I believe this paper can be interesting, as it constitutes the extension to the time domain for sparse deep learning theory. Thank you again to the authors for the detailed responses.
> >
> > # References
> > [1] Conformal prediction beyond exchangeability. The Annals of Statistics, 2023.

---

> > > ### Author Response · Authors · 2023-08-18
> > >
> > > Thank you for your continued valuable feedback! Your insights and comments have been instrumental in improving our paper. We sincerely appreciate your time and effort in reviewing our work.
> > >
> > > ## W2: Conformal Prediction Baselines
> > >
> > > We greatly appreciate and value your opinions. In response to your suggestion, we have included NexCP [1] as an additional baseline for comparison in our revised manuscript.
> > >
> > > NexCP [1] extends the original CP methods [2] to non-exchangeable data by allocating predefined and fixed weights, represented as $\\{w_i\\}^{n}\_{i=1}$, to each data point within the calibration set denoted by $\\{z\_i = (x\_i, y\_i)\\}\_{i=1}^{n}$, where $w_1,\\dots,w\_n \\in [0,1]$. These weights play a crucial role in the method. Intuitively and theoretically, higher weights $w\_i$ are assigned to data points that are considered more "trustworthy," implying they originate from a distribution closely related to the test point $z\_{n+1} = (x\_{n+1}, y\_{n+1})$. For instance, when a data point $z\_i$ corresponds to a specific time step $i$, the weights $w\_1 \\leq \\dots \\leq w\_n$ might be chosen to favor recent data, while attaching less significance to data from distant time periods.
> > >
> > > In section 4.3 of [1], it is acknowledged that the efficacy of their proposed method is influenced by the weight choices. However, they have left the optimal selection of weights, and even the quantification of optimality, for future exploration. In our application of their method to the dataset described in section 5.1, we opted to use the same weights that were employed in their experiments, specifically $w\_i = 0.99^{n+1-i}$. To ensure consistency, we followed the identical model and training procedures detailed in appendix G and section 5.1 of our paper.
> > >
> > > As a result of these additions, we have obtained updated results, which are presented in the table below. Additionally, we will provide all the relevant code for the newly introduced baselines in our revised manuscript. The code was adapted from the original implementations available in the published code bases [1,3].
> > >
> > > | Methods | Coverage | Average PI Lengths (standard deviation) | Median PI Lengths (interquartile range) |
> > > | --------------------- | --------------------- | --------------------- |--------------------- |
> > > | PA offline | 90.06 | 20.19(3.23) | 20.59(1.82) |
> > > | AgACI online | 91.05 | 23.27(14.49) | 21.38(2.62) |
> > > | ACI $\\gamma = 0.01$ online | 90.21 | 22.13(8.50) | 20.68(2.70) |
> > > | ACI $\\gamma = 0$ online | 92.89 | 24.39(9.17) | 22.99(2.91) |
> > > | EnbPI V2 online | 91.23 | 27.99(10.17) | 26.55(5.09) |
> > > | NexCP online | 91.18 | 24.41(10.40) | 22.88(2.86) |
> > >
> > > [1] Conformal prediction beyond exchangeability. The Annals of Statistics, 2023.
> > >
> > > [2] Algorithmic learning in a random world, volume 29. Springer, 2005.
> > >
> > > [3] Conformal prediction interval for dynamic time. ICML2021.

---

> > > ### Author Response · Authors · 2023-08-18
> > >
> > > ## W3: Comparison to CP methods in real-world datasets
> > >
> > > A direct comparison between conformal methods and our method might appear slightly unfair. Conformal methods are post-training inference approaches, while our method is integrated, covering both model training and inference.  In our method, hyperparameter tuning is limited to the model training phase, and the inference step is straightforward, involving the computation of the inverse of the Fisher information matrix and prediction intervals.
> > >
> > > We wish to emphasize that even for the inference part alone, conformal methods are not free from hyperparameters. For instance, when adapting conformal methods to time series data, [1] also introduces additional hyperparameters—specifically, the weights for each data point. Furthermore, [4] introduces the parameter $\gamma$ for their adaptive procedures.
> > >
> > > Additionally, it is worth noting that the performance of conformal methods can be significantly influenced by the model learned from the training data. In contrast, our sparse learning method theoretically ensures that the resulting neural networks are robust in terms of prediction.
> > >
> > > Finally, we would like to mention that we have indeed gained a significant amount of experience in hyperparameter tuning based on our examples:
> > >
> > > ### Hyperparameters for the mixture Gaussian prior:
> > >
> > > [i] $\\lambda_n$: Typically selected from the set $\\{1e-6, 1e-7\\}$, this hyperparameter exhibits minimal sensitivity within our method. Its tuning primarily involves adjusting sparsity in model sparsification tasks, if necessary.
> > >
> > > [ii] $\\sigma\_{1,n}^2$: This hyperparameter is generally chosen from $\\{0.5, 0.1, 0.05\\}$. As explained in section E of the appendix, it controls the degree of penalty in the free space, i.e., for parameters whose absolute values exceed the threshold values. For model sparsification tasks, particularly for extremely high sparsity regimes (i.e., $80\\% - 95\\%$), based on our experiments, a slightly larger value could lead to slightly better performance. Therefore, one can opt for $0.5$. However, it's worth noting that the performance of our algorithms is not significantly affected by this hyperparameter. For model selection and uncertainty quantification tasks, a value of $0.05$ or $0.1$ can be chosen without impacting the performance of our method.
> > >
> > > [iii] $(\\sigma\_{1,n}^{end})^2$: Our method exhibits low sensitivity to this hyperparameter as well, and it is typically selected from the set $\\{1e-6, 1e-7, 1e-8\\}$. A practical approach is to choose a value that closely aligns with the $(\\sigma\_{1,n}^{init})^2$ value.
> > >
> > > [iv] $(\\sigma\_{1,n}^{init})^2$: This is the first hyperparameter that can be considered as "$\\textbf{sensitive}$". For model sparsification tasks, this hyperparameter essentially determines the final sparsity. Please refer to our explanations in section E of the appendix for guidance on determining the value of this hyperparameter to achieve a specific target sparsity. For model selection and uncertainty quantification tasks, supported by both our theoretical findings (Theorem 3.8) and experimental results, this hyperparameter should be smaller for relatively larger models and larger for relatively smaller models. One can generally choose from the set $\\{1e-5, 1e-6, 1e-7\\}$.
> > >
> > > ### Hyperparameters for the prior annealing stage:
> > >
> > > Upon initial examination, it might appear that the prior annealing stage introduces an additional set of hyperparameters that require tuning, namely $c$ (temperature), $T\_1$, $T\_2$, and $T\_3$. However, the only hyperparameter that requires a modest degree of tuning is the number of training iterations that reduces $\\sigma\_{0,n}^2$ from $(\\sigma\_{0,n}^{init})^2$ to $(\\sigma\_{0,n}^{end})^2$, or equivalently, $T\_3-T\_2$. The sensitivity of this value is relatively low, as long as the count of model parameters whose absolute values lie below the current threshold (which is dependent on the present value of $\\sigma\_{0,n}^2$) remains stable—no abrupt spikes or declines.
> > >
> > > In terms of limitations, one potential concern pertains to the calculation of the inverse of the Fisher information matrix. For large-scale problems, the sparsified model could still retain a large number of non-zero parameters. In such instances, the computational feasibility of calculating the Hessian matrix, essential for prediction interval computations, might become compromised. Nonetheless, an alternative avenue exists in the form of the Bayesian approach, which circumvents the matrix inversion challenge. A concise overview of this Bayesian strategy is provided in section F of the appendix.
> > >
> > > [1] Conformal prediction beyond exchangeability. The Annals of Statistics, 2023.
> > >
> > > [4] Adaptive conformal inference under distribution shift. NeurIPS2021.

---

> > > ### Author Response · Authors · 2023-08-18
> > > **Thank you!**
> > >
> > > Thank you once again for your thoughtful feedback, and we eagerly await any further insights you might have concerning these enhancements or any other aspects of our work.

---

> > > > ### Comment · Reviewer_shpG · 2023-08-19
> > > > **Update score, and thank you!**
> > > >
> > > > I would like to kindly thanks the authors for their valuable responses and for their hard work during the rebuttal. After this discussion I feel the work, with the new updates from the rebuttal, stands as a very interesting paper.
> > > >
> > > > The new baselines for the conformal prediction greatly improve the scope of the paper. I feel interesting discussion with other researchers of the conformal prediction field can occur because of this paper.
> > > >
> > > > The authors have done a great work during this rebuttal period and because of that I decide to increase my score accordingly. I am enthusiastic about the paper and I would like to see it accepted at NeurIPS now.

---

> > > > > ### Author Response · Authors · 2023-08-19
> > > > >
> > > > > Thank you very much for your encouraging comments and for kindly raising the score. We will work hard to incorporate all the comments from the reviewers into the revised manuscript.

---

### Official Review · Reviewer_riun · 2023-07-06

**Soundness:** 3 good
**Presentation:** 2 fair
**Contribution:** 3 good
**Rating:** 4
**Confidence:** 3

**Summary:**

This paper focuses on the theoretical analysis of sparse deep learning to time series data.  Statistical propoerties of sparse RNNs are investigated including consistency and asymptotical behaviour.  The paper presents some numerical results showing that sparse deep learning outperforms existing methods in predicting uncertainty quantification for time series data, as well as in identifying the autoregressive order for time series data and large-scale model compression. The proposed method has practical implications in some fields, such as finance, healthcare, and energy.

**Strengths:**

(1) The paper addresses an important and underexplored topic by studying sparse deep learning for dependent time series data. This fills a gap in existing research, which has mostly focused on i.i.d. data.

(2) The paper presents the empirical results that show the superiority of sparse deep learning over existing methods in predicting uncertainty quantification and autoregressive order identification for time series data. This demonstrates the practical effectiveness of the proposed method.

**Weaknesses:**

(1)Some of the theorems in references are not given in the text and some symbols are confused, e.g.,
 1)Please give the concrete content of Theorem 2 from [1] that the authors cite.
 2)Please give the definition of $O_{P^*}$ on line 174.
 3)The lemma used on line 575 should be lemma S1 of section 4 in [2]. The modifications of the above weaknesses can enhance the readability of the article.

(2)There are related references that study the theoretical analysis of deep neural networks for temporally dependent observations, such as [3]. I encourage the authors to check them out and see if they should be included in the related work.

[1]Pentti Saikkonen. Dependent versions of a central limit theorem for the squared length of a sample mean. Statistics & probability letters, 1995.
[2]Yan Sun, Qifan Song, and Faming Liang. Consistent sparse deep learning: Theory and computation. Journal of the American Statistical Association, 2021.
[3]M. Ma and A. Safikhani. Theoretical analysis of deep neural networks
for temporally dependent observations, in NeurIPS, 2022.

**Questions:**

(1)The authors assume that the time series are $\alpha$-mixing process, but in fact there are many other mixing processes, such as $\beta$-mixing and $\tau$-mixing. Please explain how the $\alpha$-mixing affects the theoretical analysis of algorithms. Whether the different mixing will cause different results?

(2)Whether the data used in the experiment satisfy the $\alpha$-mixing? If yes, please provide some supporting materials. Some auto-regressive models are not $\alpha$-mixed, e.g.,\begin{equation}\label{1}
\mathbf{x}_t=f_0(\mathbf{x}_{t-1})+\epsilon_t,~t\in \mathbb{Z},
\end{equation},
where $f_0:[-K,K]^d\rightarrow[-K,K]^d$ is a Lipschitz function with Lipschitz constant $\leq$ 2 and $\epsilon_t\sim\mathcal{B}(0.5)$ [4].

(3)Please give the proof of Lemma B.1 for completeness.

(4)What is the biggest difference between the algorithm the authors proposed in section 4 and the algorithm proposed in [5]？

(5)This article is too similar to [5] and [2]. It seems that the result of this paper is an extension from i.i.d. observations to dependent data. Please clarify what are the main difficulties in the theoretical analysis for time series data compared with i.i.d. data.

[2]Yan Sun, Qifan Song, and Faming Liang. Consistent sparse deep learning: Theory and computation. Journal of the American Statistical Association,  2021.
[4]D. W. Andrews. Non-strong mixing autoregressive processes. Journal of Applied Probability, 1984.
[5]Yan Sun, Wenjun Xiong, and Faming Liang. Sparse deep learning: A new framework immune to local traps and miscalibration. Advances in Neural Information Processing Systems, 2021.


**Limitations:**

This is a theoretical work. But part of the proof is a little vague such as proof of Theorem 3.11 and Theorem 3.12. Authors should highlight the key differences  of analysis techniques  between their work and [2] [5] (instead of just using the different tools).

[2]Yan Sun, Qifan Song, and Faming Liang. Consistent sparse deep learning: Theory and Computation.
[5]Yan Sun, Wenjnd computation. Journal of the American Statistical Association,  2021.un Xiong, and Faming Liang. Sparse deep learning: A new framework immune to local traps and miscalibration. Advances in Neural Information Processing Systems, 2021.

---

> ### Author Rebuttal · Authors · 2023-08-09
>
> Thank you so much for your review. We provide a point-by-point response to your comments below.
>
> $\textbf{W1: Readability: Theorems and Lemmas}$
>
> Thank you very much for your thorough review of our paper. We sincerely appreciate your feedback, and in the camera-ready version, we will address the mentioned weaknesses as follows:
>
> (i) We will add the concrete content of Theorem 2 from [2] to the text for better clarity and reference.
>
> (ii) $O_{P^{*}}(a_n)$ is used to represents a sequence of random variables bounded in probability. Let $\\{X_n\\}$ be a sequence of random variables, and let $\\{a_n\\}$ be a sequence of strictly positive reals. We say $X_n/a_n$ is bounded in probability, if for every $\epsilon > 0$, there exists $M_\epsilon > 0$, such that
>
> $$P^{*}(|X_n/a_n| > M_\\epsilon) < \\epsilon$$
>
> for all $n$, and we can write it as $X_n = O_{P^{*}}(a_n)$. We will add this definition in the revision.
>
> (iii) You are correct; the lemma used on line 575 should indeed refer to lemma S1 of section 4 in [4]. We will make this correction in the revised version of the paper.
>
> $\textbf{W2: Related references}$
>
> Thank you for providing such useful information! We greatly appreciate your suggestion, and we will add the work [3] to the related work section for discussion. Upon careful reading of [3], we observed that their focus is on establishing consistency rates for prediction error in the context of temporally dependent observations. In contrast, our work takes a Bayesian approach, which offers distinct statistical properties, to address a similar but inherently different problem. Please also see our responses to reviewer AFD6.
>
> $\textbf{Q1: Different mixings}$
>
> Our results apply to other types of mixing sequences as implied by the following hierarchy of the five mixing conditions:
> (i) $\psi$-mixing implies $\phi$-mixing;
> (ii) $\phi$-mixing implies both $\rho$-mixing and $\beta$-mixing;
> (iii) $\rho$-mixing and $\beta$-mixing each impliy $\alpha$-mixing.
>
> See reference [1] (Theorem 3.11) for theoretical justifications for the above hierarchy:
>
> $\textbf{Q2: Data with mixing property}$
>
> Thank you very much for your suggestion. We use those datasets in order to fairly compare with current state-of-the-art methods. We conducted additional AR order selection experiments using the following exponential autoregressive order model from [5].
>
> $$y_i = \left(0.8-1.1\exp\\{-50y_{i-1}^2\\}\right)y_{i-1} + \eta_i,$$
>
> where $\eta_i \sim N(0,1)$ are i.i.d. Gaussian random noises. This model is shown to be $\alpha$-mixing according to [5], please see our new experimental details and results for this model in our responses to reviewer UrAN (W4).
>
> $\textbf{Q3: Proof of Lemma B.1}$
>
> We will give the complete proof of Lemma B.1 for completeness in the revision.
>
> $\textbf{Q4: Algorithm}$
>
> Please see our responses to reviewer AFD6 (W2).
>
> $\textbf{Q5: Novelty}$
>
> Please see our responses to reviewer AFD6 (W1).
>
> [1] R.C. Bradley (2007) Introduction to Strong Mixing Conditions. Vol. 1. Kendrick Press, Heber City (Utah).
>
> [2] Dependent versions of a central limit theorem for the squared length of a sample mean. Statistics & probability letters, 1995.
>
> [3] Theoretical analysis of deep neural networks for temporally dependent observations. NeurIPS2022.
>
> [4] Consistent sparse deep learning: Theory and computation. JASA2021.
>
> [5] Identification of nonlinear time series: First order characterization and order determination. Biometrika, 77(4):669–687, 1990.

---

### Official Review · Reviewer_UrAN · 2023-07-10

**Soundness:** 3 good
**Presentation:** 3 good
**Contribution:** 3 good
**Rating:** 7
**Confidence:** 1

**Summary:**

For iid data, sparse DL has been shown as a way towards consistency of input-output mappings and well understood distribution of model predictions. This theory is missing for time-series data however, which the authors thus introduce here. In particular, the following results for RNNs with Gaussian mixture parameters are presented:

- Consistency of posteriors, structure selection and input-output mappings
- (asymptotic) normality of predicted values

**Strengths:**

The paper was well organized and written.

Extending the recent results from sparse DL work to time-series is valuable and a novel contribution.

The consistency results presented here are convincing and valuable (at least to this reviewer who is not an expert in this field) and should pave way for a lot further work in the area -- the assumptions made in the theoretical parts are fair for this nascent area of study and provides a good starting point.


The authors show how the method can be framed as regularization in the way of laplace approximation at the MAP estimator which is useful in practice.

The experiments provide evidence of superior performance in comparison to CP methods.


**Weaknesses:**

Comparison to other possible approaches to uncertainty quantification is lacking: there is no comparisons or discussion of this method to the alternatives mentioned in the introduction: "include multi-horizon probabilistic forecasting [33], dropout-based methods [34], and recursive Bayesian methods [35].". This is the case with the experiments, but also on theoretical side. Naturally theory for the other methods may not exist in the same vain, but it would have been useful if the authors tried to make some further comparisons.

One limiting assumption here is that $\mu*{\star}$ can be approximated by a sparse RNN arbitrarily well (i.e. ground-truth sparse model).   Especially since the connectivity of this *true* RNN model is assumed to be limited. Also, as the authors point out, universal approximation does not hold fully (but can still hold for some nonlinear function classes).  Relatedly, theorem 3.8 relates to the estimation error only, but what about the approximation error?

The assumptions and conditions required for Theorems 3.8 onward are not discussed in sufficient depth.

The AR order selection experiments, while promising, are quite narrow (only 5 datasets from a single model). More different datasets would be needed with different settings (of window sizes etc.)




**Questions:**


Minor things:

There could be a bit more discussion about mixture priors: if and why they are a reasonable choice

Consider adding a comma around in line 117 to make the meaning of "only" and "or" more clear in sentence "the input $x_i$ can contain only $y_{i-1}$ or $y_{i-1:i-r}$ for some r>1". Now it's not clear if only refers to "only either of a or b", or "only a, or if not a then b is the only alternative".

I found it  confusing that $H$ was used to denote the layers and $L$ the number of hidden neurons. I would have found them more logical the other way around.

As a misc. note: I found the font of the paper odd, but that could be a local issue on my side.


**Limitations:**

In general limitations are discussed fairly well --  especially given the limited space. As always, this type of theoretical works is limited by assumptions that are made in particular. Here e.g. Gaussian mixture prior is assumed and I would be curious to hear more about the limitations of this choice of prior in more detail. What alternatives are there? The mixing assumptions can be limiting too, but the authors do acknowledge that much.

As mentioned above, it would also be nice to have bit more discussion on the assumptions and conditions of Theorems 3.8 and the other consequent theoretical results.

---

> ### Author Rebuttal · Authors · 2023-08-09
>
> Thank you so much for your review. We provide a point-by-point response to your comments below.
>
> $\textbf{W1: Comparison to other uncertainty quantification approaches}$
>
> We apologize for the confusion. We actually included MQ-RNN [1] (multi-horizon probabilistic forecasting) and DP-RNN [2] (dropout-based method) as the two baselines for the set of time series experiments. We will make sure the baselines used in the experiments are clear by adding the corresponding citations in the table.
>
> $\textbf{W2: Approximation error}$
>
> Apologies for the confusion. The convergence rate of Theorem 3.8 actually depends on the approximation error. It is essentially a sum of the approximation error and the estimation error, i.e., $\epsilon_n^2 = O(\varpi^2_n) + O(\zeta_n^2)$ in line 173 of the paper. Please also see our reply to W3Q3 to reviewer TwH7.
>
> $\textbf{W3: Discussions about Theorems 3.8}$
>
> Thank you for suggestions. We will add more discussions about the assumptions and conditions. Additional assumption for theorem 3.9 is essentially an identifiability assumption, which is commonly used for variable selection. For theorem 3.11 and 3.12, additional assumptions and discussions are given in section C in appendix.
>
> $\textbf{W4: Additional AR order selection experiments}$
>
> Thank you very much for your suggestion! As suggested, we have conducted additional AR order selection experiments with more window sizes (6 window sizes) using the following exponential autoregressive order model from [5].
>
> $$y_i = \left(0.8-1.1\exp\\{-50y_{i-1}^2\\}\right)y_{i-1} + \eta_i$$
>
> where where $\eta_i \sim N(0,1)$ are i.i.d. Gaussian random noises. We follow similar settings as section 5.2, i.e., generated 5 datasets, with the training sequence having a length of 10000, and both the validation and test sequences having a length of 1000. The results are given in the table below.
>
> |   Model   | Window size | FSR  | NSR | AR order | #hidden link | MSPE | MSFE |
> | ----------- | ----------- | ----------- | -----------| ----------- | ----------- | ----------- | ----------- |
> | PA-RNN | 1   | 0 | 0 | 1(0) | 0(0) | 1.003(0.005) | 1.004(0.004) |
> | PA-RNN | 3   | 0 | 0 | 1(0) | 0(0) | 1.006(0.005) | 0.999(0.004) |
> | PA-RNN | 5   | 0 | 0 | 1(0) | 0(0) | 1.007(0.005) | 1.000(0.004) |
> | PA-RNN | 7   | 0 | 0 | 1(0) | 0(0) | 1.006(0.005) | 1.000(0.003) |
> | PA-RNN | 10 | 0 | 0 | 1(0) | 0(0) | 1.002(0.006) | 1.002(0.004) |
> | PA-RNN | 15 | 0 | 0 | 1(0) | 0(0) | 1.002(0.007) | 1.001(0.004) |
>
> This nonlinear autoregressive order model has an AR order of 1. We use the same hyperparameters for all window sizes, and our method managed to achieve perfect results for all window sizes with respect to all model selection metrics. This might be because this is a relatively simpler model compared with the one we used in Section 5.2. We will add both the results and details of these additional experimental results in the revised version.
>
> $\textbf{Q1: Discussion about mixture priors}$
>
> The mixture Gaussian prior can be viewed as a continuous relaxation of the spike-and-slab prior, which puts a point mass at 0. Compared to the spike-and-slab prior, a significant advantage of the mixture Gaussian prior is that it is differentiable, allowing stochastic gradient-based algorithm to be used for computation. And the mixture allows us to put enough weight on the neighborhood of 0, which essentially allows the prior to satisfy conditions (a)-(c) in section B of the appendix.
>
> $\textbf{Q2: Line 117}$
>
> Thank you, we will add a comma in the revised version to clarify the meaning as you suggested.
>
> $\textbf{Q3: Notation H}$
>
> We sincerely apologize for the confusion. We mainly followed the notations used in [3,4], but we will make sure it is more clear in the revised version.
>
> $\textbf{Q4: Font of the paper}$
>
> We will have this issue addressed in the revision.
>
> [1] A multi-horizon quantile recurrent forecaster. arXiv preprint, 2017.
>
> [2] Dropout as a bayesian approximation: Representing model uncertainty in deep learning. ICML2016.
>
> [3] Consistent sparse deep learning: Theory and computation. JASA2021.
>
> [4] Sparse deep learning: A new framework immune to local traps and miscalibration. NeurIPS2021.
>
> [5] Identification of nonlinear time series: First order characterization and order determination. Biometrika, 77(4):669–687, 1990.

---

> > ### Comment · Reviewer_UrAN · 2023-08-16
> >
> > Thank you for your answers -- I think they are reasonable and I would gladly see the paper published. Reading through all the reviews and rebuttals it is evident to me that I don't possess sufficient understanding of the mathematical details and relevant literature to fairly judge those aspects. Thus I have lowered the confidence of my review from 2 to 1.

---

### Official Review · Reviewer_TwH7 · 2023-07-11

**Soundness:** 3 good
**Presentation:** 2 fair
**Contribution:** 3 good
**Rating:** 7
**Confidence:** 2

**Summary:**

This paper proposes to extend sparse deep learning in the context of time series data. This context is different from the classical one as samples are not *i.i.d.* but dependent. The paper shows theoretical results -- posterior consistency and asymptotic normality of the weights --, a computation method based on posterior annealing and some numerical experiments that showcase the performance of the method on real datasets.

**Strengths:**

- The paper extend sparse deep learning result to the challenging context of dependent data, applicable for time series.
- The many reported results show improvement of the method compared to existing state-of-the-art methods and seems to yield interpretable models for time series.

**Weaknesses:**

- It is not totally clear what is really different between the results in [17] and the results presented here (see **Q1**).
- The writing is not always clear (**Q2**), in particular for non-expert in sparse deep learning. It is not clear why `Eq.(4)` admits an non-empty set of minimizers (**Q3**).
- There are many hyperparameters in the methods and while there value is reported, it is not clear how to select them (**Q4**).

**Questions:**

- **Q1** - The theoretical results are very similar to the ones of [17], which considers *iid* data with DNN. As the proposed method consider the input of the network as fixed segments $x_{i:i-M_l+1}$, the considered networks can be assimilated as weight-tied DNN, so the class of architecture is not really different. And in `Theorem3.8`, it is not clear where the dependence between the samples intervenes. In particular, I am surprised that we get similar sample efficiency rates between the two settings, and I wonder what is lost by dropping the iid assumption. Could the authors comment on this point?
- **Q2** -  Many details are deferred to the appendix or referred to in other articles. This makes the paper sometimes hard to read. In particular, `Section.4` is very hard to follow, in particular the part on the transformation of the Bayesian method into a regularization method. As this constitutes the core of the proposed method, I think this could be better explained in the main article.
- **Q3** - In equation (4), it is unclear what are the conditions for  $\varpi_n$ so that the argmin is not empty. In particular, as the size of the neural network in the set $\mathcal G_n$ is constrained in `Assumption3.4`. This specific assumption that there exists a minimizer is used in Eq. bellow l.657, which relies on the fact that for all $n$, $\|\mu(X, \beta, \gamma) - \mu^*\|_{L^2(\Omega)} \le \varpi_n$ is not empty. This is not guaranteed I think, in particular as $\varpi_n$ goes to 0 at the same rate as the network size increases, while the data point to interpolate increases much faster. Note that a similar question arises in the proof for [17]. It seems that the results in [17] show that universal approximation holds for a specific class of functions even for sparse DNN, but it is unclear if the constraints in `Assumption3.4` allows for this. But maybe this is due to a misunderstanding on my side.
- **Q4** The method relies on many hyperparameters, that are specified in the appendix. However, the process for their selection is not reported, and it would be nice to understand how sensitive the hyperparameter selection of the model is.

**Minor remarks and typos:**
- **m1** - `Section 5.2` - Isn't it possible to estimate the AR order of window size =1 by looking at the average dependency length? Or at least estimating if changing one sample changes the result?
- **m2** - The proposed results seem to relate to the compress sensing and sparse coding theory, it would be nice to draw the connection between these two fields in the introduction. But I have no specific results in mind, so maybe it is too far-fetched.

*unnumbered remarks don't call for answers*
- Eq. (2) - Missing $+ v^{H_n}z^{H_n}_{i-1}$ at the end to show the structure that is not just feedforward?
- l.123 - $\gamma = \{\gamma_i \in \{0, 1\} : j=1\dots K_n\}$ -> $\gamma \in \{0, 1\}^{K_n}$?
- l.155 - an RNN of size ... has been" -> "... is".
- l.192 - mismatch between name and acronym `MIPP` and `marginal posterior inclusion probability`.
- l.218 - $H_n$ is already the number of layers (l.113), finding a different notation would help the reader follow the arguments.
- l305 - "In particular, Our method" -> "our".

**Limitations:**

* see weaknesses.

---

> ### Author Rebuttal · Authors · 2023-08-09
>
> Thank you so much for your review. We provide a point-by-point response to your comments below.
>
> $\textbf{W1 and Q1: Novelty}$
>
> The seminal work [1] established a general theoretical framework for studying the asymptotic behavior of posterior distributions and Bayesian estimators for high-dimensional statistical models. The posterior consistency theory established therein requires three conditions for the models and the prior, namely: (a) The set of models considered in the analysis cannot be too large (in terms of packing numbers).
> (b) The set of models almost encompasses the support of the prior distribution.
> (c) The prior distribution places a sufficient amount of mass near the true model.
> The work [2] further extended this theory to non-i.i.d data under the three general conditions, leading to a similar posterior contraction rate. Dropping the i.i.d assumptions disables many inequalities used for the i.i.d case, necessitating specific considerations. However, the resulting contraction rates are not necessarily very different. In the same vein, we get a similar posterior contraction rate as reference [4] (cited in the paper).  Please also see our responses to reviewer AFD6 (W1)
>
> $\textbf{W2 and Q2: Transformation of the Bayesian method}$
>
> In the revision, we will add a lemma regarding applications of the Laplace approximation to Bayesian neural networks. Similar to Theorem 2.3 in [4], the main idea is to connect posterior consistency to the consistency of the maximum a posteriori (MAP) estimator. Then the prior annealing algorithm is employed for finding the optimum of the posterior.
>
> $\textbf{W3 and Q3: Equation 4 and approximation error}$
>
> This is a very thoughtful question. Lemma 4.1 of [3], which, through the trick of independent block sequence construction, shows that many properties of the i.i.d processes can be extended to mixing processes. While the lemma was proven for the case of $\beta$-mixing, the author did mention her doubts about its applicability to $\alpha$-mixing. Therefore, at least for the sequences of $\beta$-mixing (which implies $\alpha$-mixing or can be viewed as a subclass of $\alpha$-mixing), the non-empty of the sparse RNN set can be guaranteed for many classes of functions. This issue will be mentioned in the revision.  Moreover, the contraction rate of our posterior consistency results is essentially a sum of the approximation error and estimation error, i.e. $\epsilon_n^2 = O(\varpi^2_n) + O(\zeta_n^2)$ in Theorem 3.8, which provides the flexibility to be combined with other approximation theories to give get exact order of $\epsilon_n^2$.
>
> $\textbf{W4 and Q4: Hyperparameters}$
>
> Thank you for your feedback. Regarding the four hyperparameters related to the mixture Gaussian priors $(\lambda_n, \sigma_{1,n}, \sigma_{0,n}^{init}, \sigma_{0,n}^{end})$, our algorithm is not sensitive to $\lambda_n$ and $\sigma_{0,n}^{end}$. $\lambda_n$ is only used for adjusting the target sparsity if needed, and it is usually selected from {1e-6, 1e-7}. $\sigma_{0,n}^{end}$ is generally set between {1e-5, 1e-6, 1e-7}, with the exact value depending on $\sigma_{0,n}^{init}$. We gradually decrease $\sigma_{0,n}^{init}$ to $\sigma_{0,n}^{end}$ during prior annealing. Based on the number of iterations/epochs and the learning rate used in this stage, the difference between $\sigma_{0,n}^{end}$ and $\sigma_{0,n}^{init}$ should be adjusted so that sparsity remains stable.
> Selecting $\sigma_{0,n}^{init}$ depends on the specific task. For example, as explained in Section E of the appendix, we select $\sigma_{0,n}^{init}$ by achieving the target sparsity for model sparsification tasks. For other tasks like model selection and uncertainty quantification, $\sigma_{0,n}^{init}$ is not as sensitive as long as sparsity is stable during annealing. Our algorithm is also robust to the temperature hyperparameter. We will add a section in the appendix instructing hyperparameter selection in the revised version.
>
> $\textbf{Minors 1:}$
>
> Thank you for your feedback. Through our selected network structure, the AR order will not be directly available in the case
> window size=1. However, in practice, it is still possible to estimate the AR order by looking at the average dependency length. This point will be mentioned in the revision.
>
> $\textbf{Minors 2:}$
>
> Thank you very much for your valuable suggestions! We will review compress sensing and sparse coding theory, and add them accordingly in the introduction.
>
> $\textbf{Typos}$
>
> Thank you for your feedback. We will fix these typos in the revised version.
>
>
> [1] Convergence rates of posterior distributions. Annals of Statistics, 2000.
>
> [2] Convergence rates of posterior distributions for non-i.i.d. observations. Ann. Statist. 35(1), 192–223
>
> [3] Rates of convergence for empirical processes of stationary mixing sequences. The Annals of Probability, 22, 94-116.
>
> [4] Consistent sparse deep learning: Theory and computation. JASA2021.

---

> > ### Comment · Reviewer_TwH7 · 2023-08-19
> > **Answer to authors' rebutal**
> >
> > I thank the authors for the detailed answers to my review. I have read the review by other reviewers. Overall, I think this is an interesting paper that is worth accepting at the conference therefore I will upgrade my rating to 7. I particularly appreciated the comment about the hybridation of algorithm vs data modeling, which I think is worth mentioning in the intro. Below are further comments about the answers.
> >
> > **Q1** - Thanks for the explanation. I think adding an explicit mention of this point in the manuscript will help understand the theoretical contribution of the paper.
> >
> > **Q3** - I am a bit lost by your answer but probably due to a lack of knowledge about $\alpha/\beta$-mixing. From what I understand, the non-emptiness can be ensured at least for $\beta$-mixing by constructing independent blocks and transferring the result from [17].

---

> > > ### Author Response · Authors · 2023-08-19
> > >
> > > Thank you very much for your encouraging comments and for kindly raising the score.
> > >
> > > Q1R: As suggested, the comment about the hybridization of algorithm versus data modeling will be incorporated into the introduction of the revised manuscript.
> > >
> > > Q3R: Yes, your understanding is correct. The non-emptiness can be ensured at least for $\beta$-mixing by constructing independent blocks and transferring the result from [17].

---

### Official Review · Reviewer_AFD6 · 2023-07-19

**Soundness:** 2 fair
**Presentation:** 3 good
**Contribution:** 3 good
**Rating:** 5
**Confidence:** 2

**Summary:**

The paper extends the sparse DNN theory from [1, 2] to time series data and proposes the sparsity method for RNNs. This method aims to improve uncertainty quantification on time-series tasks and provide state-of-the-art RNN compression.

**Strengths:**

* The combination of DNN sparsification and uncertainty quantification for time series data is an interesting direction.
* The theoretical groundwork seems complete and convincing.

**Weaknesses:**

**Novelty**

* **Theory** - The theoretical results for RNNs and time series seem almost identical to [1, 2] aside from minor time series specifics. I appreciate that the authors explicitly discuss the connection to these works. However, I have concerns if the theoretical contribution is significant enough.
* **Computation** - the prior annealing algorithm (PA) for model sparsification is identical to [2]. Constructing the prediction intervals for time series might be novel, but it is a minor contribution from my perspective.

**Large-scale model compression**

The authors state that one of their contributions is s.o.t.a. large-scale RNN compression method.

1. In my opinion, the scope of the model compression area is much larger than the weight sparsification. One should either reformulate "compression" to "weight sparsification" or compare with low-rank approximation, quantization, and knowledge distillation methods.
2. The proposed method is compared only with AGP[5] on PTB. The authors state that this area has problems with benchmarks and baselines. I agree that it makes the fair comparison challenging. On the other hand, I think the results on a single dataset against a single baseline from 2017 are not sufficient to support such a strong claim. I would consider at least one more dataset and compare with, e.g., GraNet[6], and some sparsification methods for RNNs: e.g., [7, 12]. Otherwise, please revise the claims and contributions.
3. (Minor) All experiments are entirely deferred to the appendix. If the authors declared this as one of the contributions, I would expect it in the main paper.

**Related work**

* **Uncertainty quantification:** please add DropConnect[3] and deep ensembles[4] as uncertainty estimation methods and consider them for comparison in the multi-horizon setting.
* **Sparse DNN** should be largely extended by discussing different before-training [8], post-training [9] and during-training [5,6,7,10,11] approaches. Regarding the RNN sparsification, I would consider citing and discussing the following works [12, 13, 14].

**Clarity**

It was hard to follow some theoretical parts of the paper due to large equations and overloaded notation, e.g., Theorem 3.8. I can see that the authors put effort into simplifying the notation and supporting the theory with discussions in some places. Nevertheless, I think it would be great to do an extra pass to make reading more clear.

(Minor) Figure 1 takes a lot of space but just aims to explain M_l and R_l notation and the RNN setting. I would suggest making it more informative or maybe consider moving it to the appendix. The model compression results arguably seem more important than this figure.

---

[1] Consistent sparse deep learning: Theory and computation. JASA, 2021

[2] Sparse Deep Learning: A New Framework Immune to Local Traps and Miscalibration, NeurIPS 2021

[3] Dropconnect is effective in modeling uncertainty of bayesian deep networks 2019

[4] Simple and scalable predictive uncertainty estimation using deep ensembles NIPS2017

[5] To prune, or not to prune: exploring the efficacy of pruning for model compression, 2017

[6] Sparse training via boosting pruning plasticity with neuroregeneration, NeurIPS2021

[7] Bayesian Compression for Natural Language Processing, EMNLP2018

[8] SNIP: Single-shot network pruning based on connection sensitivity, ICLR2019

[9] The Lottery Ticket Hypothesis: Finding Sparse, Trainable Neural Networks, ICLR2019

[10] Learning Sparse Neural Networks through L0 Regularization, ICLR2018

[11] Scalable training of artificial neural networks with adaptive sparse connectivity inspired by network science, Nature 2018

[12] Spectral Pruning for Recurrent Neural Networks, AISTATS2020

[13] Structured Sparsification of Gated Recurrent Neural Networks, AAAI2020

[14] Stage-Wise Magnitude-Based Pruning for Recurrent Neural Networks. TNNLS2022.

**Questions:**

1. Could the authors address the novelty and model compression evaluation concerns?
2. In my understanding, deep ensembles are s.o.t.a. for uncertainty estimation. Is it a reasonable baseline in the multi-horizon setting?

**Limitations:**

The authors addressed the limitations to some extent. One can also discuss more limitations of the proposed algorithm.

---

> ### Author Rebuttal · Authors · 2023-08-09
>
> Thank you so much for your review. We provide a point-by-point response to your comments below.
>
> $\textbf{W1: Novelty}$
>
> The seminal work [1] has established a general theoretical framework for studying the asymptotic behavior of posterior distributions and Bayesian estimators for high-dimensional statistical models. The posterior consistency theory established therein requires three conditions for the models and the prior given in section B of the appendix, lines 633-636. In words, these conditions require:
> (a) The set of models considered in the analysis cannot be too large (in terms of packing numbers).
> (b) The set of models almost encompasses the support of the prior distribution.
> (c) The prior distribution places a sufficient amount of mass near the true model.
> However, verifying conditions (a)-(c) for a specific problem is not trivial. For example, [2] addressed the issue for high-dimensional generalized linear models; [3] tackled the issue for deep neural networks with i.i.d. data; and [4] dealt with the issue for nonparametric Bayesian models. In this paper, we address the issue for RNNs with $\alpha$-mixing sequences.
> Although we employ similar conditions as in [3] to constrain the set of neural networks used in data modeling, verification of conditions (a)-(c) remains non-trivial. This is detailed in the Appendix of the paper. Notably, some inequalities used for i.i.d. data no longer hold, such as the inequalities on $d_t$ difference (a generalization of the Kullback-Leibler divergence) used for proving Proposition 1 in [2]. To our knowledge, this paper provides the first theoretical study of sparse RNNs for time series data from the perspective of structure selection, parameter estimation, and prediction uncertainty quantification. Given the wide usefulness of RNNs for time series data [11] , we believe that their theory deserves special attention.
>
> $\textbf{W2: Algorithm}$
>
> Yes, the prior annealing algorithm is a direct application of the algorithm in [5]. Our main contribution is extending the theoretical results to RNN models for time series data. Additionally, our experiments demonstrate the advantages of the proposed method over prior works.
>
> $\textbf{W3: Model sparsification and baselines}$
>
> Thank you for the suggestion. In the revision, we will change the term `compression` to `sparsification` as suggested. We have conducted an additional experiment comparing our model sparsification approach to the baseline method proposed in [6]. Following their experimental setup exactly, we trained a 1 layer RNN with 128 hidden units on PTB data, using the same batch size, number of epochs, and 5 independent runs. As shown in table below, our approach achieves better test perplexity than [6] under similar (and higher) sparsity levels. All baseline results are directly adopted from their paper.
>
> |   Methods   | Test Perplexity  | Sparsity |
> | ----------- | ----------- | ----------- |
> | Baseline     |114.66 (0.35)  | 0%  |
> | $\textbf{PA (ours)}$  | 117.80 (0.10)  | 70%  |
> | Spectral w/ rec [6] | 124.26 (0.39)  | 67%  |
>
> We will add both the results and details of these additional experimental results in the revised version.
>
> $\textbf{W4: Related work: uncertainty quantification}$
>
> For the multi-horizon forecasting experiments, we followed the setup from [7] and used an LSTM as the base prediction model. Per your suggestion, we will add discussion of deep ensembles [8] and DropConnect [9] to the related work section. Those methods were originally developed for i.i.d. data like image classification. Therefore, adapting them to RNNs for multi-horzion time series forecasting would require careful tuning and an extension of their original methods with the Bonferroni correction to ensure a fair comparison.
>
> $\textbf{W5: Related work: sparse DNN}$
>
> The lottery ticket hypothesis [10] shows that for many vision tasks, there exist sparse networks that can be trained from scratch to achieve good performance, but the models must have special initialization. The work shed light on research on pruning before training. During training approaches typically add regularization during training to force parameters to go to 0. Post-training approaches operate on trained neural networks and attempt to remove network parameters based on some pruning criteria such as parameter magnitude, Hessian of the loss function, etc. We will add discussion of these works in the revision. From this perspective, our work is mostly aligned with the during-training approach.
>
> $\textbf{W6: Clarity}$
>
> In the revision, we will continue our efforts to simplify the notation and provide additional discussions for the theory to enhance the readability of the paper.
>
> $\textbf{W7: Figure 1}$
>
> In the revision, we will move Figure 1 to the appendix and add more model compression results to the main body of the paper.
>
> [1] Convergence rates of posterior distributions. Annals of Statistics, 2000.
>
> [2] Bayesian variable selection for high dimensional generalized linear models: convergence rates of the fitted densities. The Annals of Statistics, 2007.
>
> [3] Consistent sparse deep learning: Theory and computation. JASA2021.
>
> [4] Nonparametric bayesian model selection and averaging. 2008.
>
> [5] Sparse deep learning: A new framework immune to local traps and miscalibration. NeurIPS2021.
>
> [6] Spectral pruning for recurrent neural networks. AISTATS2020.
>
> [7] Conformal time-series forecasting. NeurIPS2021.
>
> [8] Simple and scalable predictive uncertainty estimation using deep ensembles. NeurIPS2017.
>
> [9] Dropconnect is effective in modeling uncertainty of bayesian deep networks. Scientific reports, 2021.
>
> [10] The Lottery Ticket Hypothesis: Finding Sparse, Trainable Neural Networks, ICLR2019.
>
> [11] Deep learning for twelve hour precipitation forecasts. Nature communications, 13(1):1–10, 2022.

---

> > ### Comment · Reviewer_AFD6 · 2023-08-17
> > **Response to rebuttal**
> >
> > I would like to thank the authors for their thoughtful clarifications and additional results. I’ve also read other reviews and responses and, as a result, got a bit more optimistic about the submission.
> >
> > **Novelty** Thanks for clarifications about the novelty of the theoretical contribution. As a practitioner, I struggle to judge if verifying similar conditions for the dependent data is sufficient for the main contribution. Thus, I would like to see the opinion of Reviewer riun, who also raised a similar concern.
> >
> > **Model sparsification** Thanks for the comparison with spectral pruning. I believe it makes the comparison slightly better. Having more datasets and methods for comparison would still be great, but I do not think it is a big problem since the main contribution is theoretical.
> >
> > Overall, I do not have strong objections to acceptance if other reviewers confirm the significance of the theoretical contribution against [1, 2]. If accepted, the paper should be largely revised to address reviewers’ suggestions and misunderstandings. I keep my score for now and will consider raising it after the discussions.
> >
> > [1] Sparse deep learning: A new framework immune to local traps and miscalibration. NeurIPS2021
> >
> > [2] Consistent sparse deep learning: Theory and computation. JASA2021

---

> > > ### Author Response · Authors · 2023-08-18
> > >
> > > Thank you for your encouraging words. Of course, we will fully take into account the comments and suggestions provided by the reviewers during the revision process.
> > >
> > > *Question about the novelty.*
> > >
> > > *Reply:*  Other than our clarifications about the novelty of the theoretical contribution, we also want to elaborate our contribution in a broader context of statistical modeling. As discussed in [1], two distinct cultures exist for statistical modeling: the 'data modeling culture' and the 'algorithmic modeling culture'. The former focuses on simple generative models that explain the data, potentially lacking a consistent estimate of the true data-generating mechanism due to the model's inherent simplicity. The latter, on the other hand, aims to find models that can predict the data regardless of complexity.
> > >
> > > Our proposed method occupies a middle ground between these two cultures. It seeks to identify a parsimonious model within the realm of complex models, while also ensuring a consistent estimation of the true data-generating mechanism. From this perspective, our work and [2,3] represent a new culture as a hybridization of the 'algorithmic modeling culture' and the 'data modeling culture'. This hybridization holds the potential to expedite advancements in modern data science. To illustrate, an increasing number of authors have recently begun exploring ways to sparsify LLM models. In our limited experience, our method has demonstrated efficacy in this context as well.
> > >
> > > [1] L. Breiman. Statistical modeling: The two cultures (with comments and a rejoinder by the author). Statistical Science, 16:199–231, 2001.
> > >
> > > [2] Y. Sun, W. Xiong, and F. Liang (2021) Sparse deep learning: A new framework immune to local traps and miscalibration. NeurIPS2021
> > >
> > > [2] Y. Sun, Q. Song and F. Liang (2021) Consistent sparse deep learning: Theory and computation. JASA2021.

---

> > > > ### Comment · Reviewer_AFD6 · 2023-08-21
> > > > **Official Comment by Reviewer AFD6**
> > > >
> > > > Thank you for the additional interesting comments! I’ve read the discussions one more time and decided to update the score, marginally leaning toward acceptance. I still believe that theoretical contribution is somewhat incremental and encourage the authors to go beyond adapting the same results to the new data types or architectures in the future.

---

> > > > > ### Author Response · Authors · 2023-08-22
> > > > >
> > > > > Thank you very much for your encouraging comments and for kindly raising the score. As suggested, we will certainly extend the proposed method further to new data types or architectures in the future, as a continuation of the development for the hybridized culture of stochastic modeling.

---

### Comment · Area_Chair_rSk3 · 2023-08-18
**Acknowledging author rebuttals**

Dear all,

I want to thank the authors for their rebuttals and want to acknowledge that these will be taken into account.
Unfortunately, 2 reviewers have still not replied to the author rebuttal and I explicitly urge them (again) to please reply to the rebuttals as soon as possible since the author/reviewer discussion period ends soon.

There are still discrepancies in the scores this submission has received so far and the reviewer-author discussion is crucial to clarify the different viewpoints.

Best regards

---

### Decision · Program_Chairs · 2023-09-21

**Decision:**

Accept (poster)

**Comment:**

There was initially quite some uncertainty around the contribution of this submission and reviewers struggled with the limited clarity of the original exposition (partly also due to complex notation, etc). We strongly encourage the authors to work on the clarity of the exposition as suggested by the reviewers for the final version. The authors went through great length to generate additional results and provide clarifications in the rebuttal. We strongly encourage the authors to include these clarifications and additional results in the final version. After some back and forth, the reviewers were largely convinced (albeit still with relatively low certainty) of the relevance and novelty of this submission and overall I think it is an adequate contribution to NeurIPS.